# Near-Optimal Multi-Perturbation Experimental Design for Causal Structure Learning

**Scott Sussex**
Department of Computer Science
ETH Zürich
Zürich, Switzerland
`scott.sussex@inf.ethz.ch`

**Andreas Krause**
Department of Computer Science
ETH Zürich
Zürich, Switzerland

**Caroline Uhler**
Laboratory for Information & Decision Systems
Massachusetts Institute of Technology
Cambridge, MA

## Abstract

Causal structure learning is a key problem in many domains. Causal structures can be learnt by performing experiments on the system of interest. We address the largely unexplored problem of designing a batch of experiments that each *simultaneously intervene on multiple variables*. While potentially more informative than the commonly considered single-variable interventions, selecting such interventions is algorithmically much more challenging, due to the doubly-exponential combinatorial search space over sets of composite interventions. In this paper, we develop efficient algorithms for optimizing different objective functions quantifying the informativeness of a budget-constrained batch of experiments. By establishing novel submodularity properties of these objectives, we provide approximation guarantees for our algorithms. Our algorithms empirically perform superior to both random interventions and algorithms that only select single-variable interventions.

## 1 Introduction

The problem of finding the causal relationships between a set of variables is ubiquitous throughout the sciences. For example, scientists are interested in reconstructing gene regulatory networks (GRNs) of biological cells [11]. Directed Acyclic Graphs (DAGs) are a natural way to represent causal structures, with a directed edge from variable $X$ to $Y$ representing $X$ being a direct cause of $Y$ [33].

Learning the causal structure of a set of variables is fundamentally difficult. With only observational data, in general we can only identify the true DAG up to a set of DAGs called its *Markov Equivalence Class (MEC)* [35]. Empirically, for sparse DAGs the size of the MEC grows exponentially in the number of nodes [17]. Identifiability can be improved by intervening on variables, meaning one perturbs a subset of the variables and then observes more samples from the system [8, 16, 40]. There exist various inference algorithms for learning causal structures from a combination of observational and interventional data [16, 36, 40, 28, 34]. Here we focus on the identification of DAGs that have no unobserved confounding variables.

Performing experiments is often expensive, however. Thus, we are interested in learning as much about the causal structure as possible given some constraints on the interventions. In this work, we focus on the *batched setting*, where several interventions are performed in parallel. This is a natural setting in scientific domains like reconstructing GRNs. Existing works propose meaningful objective

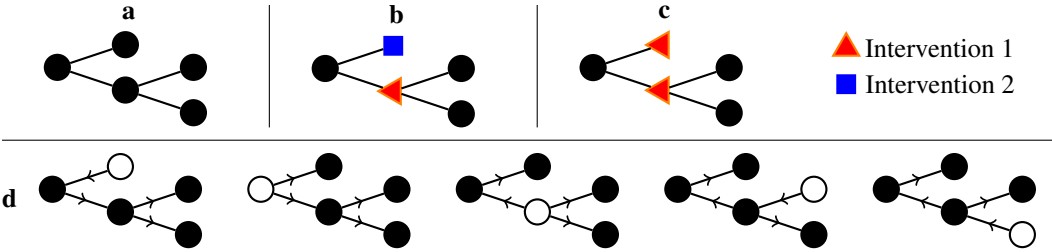

Figure 1: **a)** We illustrate the MEC of a tree graph on 5 nodes. **b)** Two single-node interventions are required to fully identify the true DAG. **c)** Only one two-node intervention is required to fully identify the true DAG. **d)** This MEC contains 5 DAGs, each corresponding to a different root node (marked white). This is a property particular to tree MECs.

functions for this batched causal structure learning problem and then give algorithms that have provable guarantees [3, 13]. However, these works focus on the setting where only a *single* random variable is perturbed per intervention. It is an open question as to whether there exist efficient algorithms for the *multiple-perturbation* setting, where more than one variable is perturbed in each intervention.

For the task of reconstructing GRNs, it is now possible for experimenters to perturb multiple genes in a single cell [2, 7]. Figures 1 b) and c) illustrate a specific example where a two-node intervention completely identifies a DAG in half as many interventions as single-node interventions. In general, it is possible for a set of $q$-node interventions to orient *up to $q$-times more* edges in a DAG than single-node interventions (see the supplementary material for a more general example). While multi-perturbation interventions can be more informative, designing them is algorithmically challenging because it leads to an *exponentially* larger search space: any algorithm must now select a set of sets.

Our main contribution is to provide efficient algorithms for different objective functions with accompanying performance bounds. We demonstrate empirically on both synthetic and GRN graphs that our algorithms result in greater identifiability than existing approaches that do not make use of multiple perturbations [13, 40], as well as a random strategy.

We begin by introducing the notation and the objective functions considered in this work in Section 2, before reviewing related work in Section 3. In Section 4 we present our algorithms along with proofs of their performance guarantees. Finally, in Section 5 we demonstrate the superior empirical performance of our method over existing baselines on both synthetic networks and on data generated from models of real GRNs.

## 2 Background and Problem Statement

**Causal DAGs** Consider a causal DAG $G = ([p], E)$ where $[p] := \{1, ..., p\}$ is a set of nodes and $E$ is a set of directed edges. Let $(i, j) \in E$ iff there is an edge from node $i$ to node $j$. Each node $i$ is associated with a random variable $X_i$. In the GRN example, $X_i$ would be the measurement of the gene expression level for gene $i$. An edge from $i \to j$ would represent gene $i$ having a causal effect on the expression of gene $j$. The functional dependence of a random variable on its parents can be described by a *structural equation model* (SEM).

The probability distribution over $X = (X_1, \ldots, X_p)$ is related to $G$ by the Markov property, meaning each variable $X_i$ is conditionally independent of its non-descendants given its parents [33]. From conditional independence tests one can determine the MEC of $G$, a set of DAGs with the same conditional independancies between variables. All members of the MEC share the same undirected skeleton and colliders [35]. A collider is a pair of edges $(i, k), (j, k) \in E$ such that $(i, j), (j, i) \notin E$. The *essential graph* of $G$, Ess($G$), is a partially directed graph, with directed edges where all members of the MEC share the same edge direction, and with undirected edges otherwise [5]. Ess($G$) uniquely represents the MEC of $G$. These MECs can be large, so we seek to perform interventions on the nodes to reduce the MEC to a smaller set of possible DAGs.

**Interventions** We use the term *intervention* to refer to a set $I \subset [p]$ of perturbation targets (variables). We assume all interventions are *hard interventions*, meaning intervening on a set $I$ removes the incoming edges to the random variables $X_I := (X_i)_{i \in I}$ and sets their joint distribution

to some interventional distribution $\mathcal{P}^I$ [9]. In the GRN reconstruction example this corresponds to, for example, running an experiment where we knockout all genes in set $I$. Some of our results extend easily to the alternative model of *soft* interventions [25], as we discuss in the supplementary material.

We use $\mathcal{I} = 2^{[p]}$ to refer to the set of all possible interventions. Our goal will be to select a batch $\xi$ of interventions, where $\xi$ is a multiset of some $I \in \mathcal{I}$. For practical reasons, we typically have *constraints* on the number of interventions, i.e., $|\xi| \leq m$ and on the number of variables involved in each intervention $|I| \leq q, \forall I \in \xi$. Namely, there are at most $m$ interventions per batch and each intervention contains at most $q$ nodes. A constraint on the number of perturbations per intervention is natural in reconstructing GRNs, since perturbing too many genes in one cell will leave it unlikely to survive. We refer to the set of $\xi$s satisfying these constraints as $C_{m,q}$. The observational distribution (no intervention) is given by $\xi = \emptyset$.

For any set of interventions $\xi$ and DAG $G$, there is a set of $\xi$-Markov equivalent DAGs. These are the set of DAGs that have the same set of conditional independencies under all $I \in \xi$ and under the observational distribution. This set of DAGs is no larger than the MEC of $G$ and can similarly be characterized by an essential graph $\text{Ess}^\xi(G)$ [16].

We will always assume that there exist no unobserved common causes of any pair of nodes in $G$. We also assume that the distribution of the random variables satisfies faithfulness with respect to $G$ [33].

**Choosing optimal interventions** We seek to maximize an objective function $F$ that quantifies our certainty in the true DAG. In general our goal is to determine

$$\arg\max_{\xi \in C_{m,q}} F(\xi).$$

A natural choice for $F$ is given by Agrawal et al. [3]. They assume that there exist parameters $\theta$ that determine the functional relationships between random variables. For example, this could be the coefficients in a linear model. Given existing data $D$, we try to choose $\xi$ that maximizes

**Objective 1 (Mutual Information (MI)).**

$$F_{\text{MI}}(\xi) = \mathbb{E}_{G|D}\mathbb{E}_{y|G,\hat{\theta},\xi}\left[\tilde{U}_{M.I}(y, \xi; D)\right], \tag{1}$$

where $y$ is the set of samples from the interventions, $\hat{\theta}$ is the current estimate of the parameters given $D$ and $G$, and $\tilde{U}_{M.I}(y, \xi; D)$ is the mutual information between the posterior over $G$ and the samples $y$. Each intervention produces one sample in $y$. The use of mutual information means the objective aims to, in expectation over all observed samples and true DAGs, minimize the entropy of the posterior distribution over DAGs. There already exist a number of algorithms for determining the posterior over DAGs from observational or experimental data [40, 36, 16].

**Infinite sample objectives** Finding algorithms that optimize the MI objective is difficult because we have to account for noisy observations and limited samples. To remove this complexity, we study the limiting case of *infinitely many samples* per unique intervention. The constraints given by $C_{m,q}$ still stipulate that there can be only $m$ unique interventions, but each intervention can be performed with an infinite number of samples. We also assume that an essential graph is already known (i.e., we have infinite observational samples and infinite samples for any experiments performed so far). For objectives with infinite samples per intervention, we treat $\xi$ as a set of interventions, not a multiset, since there is no change in objective value for choosing an intervention twice. Consider $\xi'$ to be the set of interventions contained in our dataset before our current batch. In this setting, maximizing Objective 1 reduces to maximizing

**Objective 2 (Mutual info. inf. samples (MI-$\infty$)).**

$$F_\infty(\xi) = -\frac{1}{|\mathcal{G}|} \sum_{G \in \mathcal{G}} \log_2 |\text{Ess}^{\xi \cup \xi'}(G)|, \tag{2}$$

where $\text{Ess}^{\xi \cup \xi'}(G)$ refers to the updated essential graph after performing interventions in $\xi$. The objective aims to, on average across possible true DAGs, minimize the $\log$ of the essential graph size after performing the interventions. The derivation of this objective is given in the supplementary material.

Ghassami et al. [13] study a different objective in the infinite-sample setting. The objective seeks to, on average across possible $G$ given the current essential graph, orient as many edges as possible. Let

$R(\xi, G, \xi')$ be the set of edges oriented by $\xi$ if the true DAG is $G$, and the essential graph is given by the $\xi'$-MEC.

**Objective 3 (Edge-orientation (EO)).**

$$F_{\text{EO}}(\xi) = \frac{1}{|\mathcal{G}|} \sum_{G \in \mathcal{G}} |R(\xi, G, \xi')|. \tag{3}$$

The function $R$ is computed as follows. Firstly, $\forall I \in \xi$, orient undirected edge $i - j$ in $\text{Ess}^{\xi'}(G)$ if $i \in I$ but $j \notin I$ or vice-versa. Secondly, execute the *Meek Rules* [26], which allow inferring additional edge orientations (discussed in the supplementary material) on the resulting partially directed graph. Finally, output the set of all edges oriented. Agrawal et al. [3] show that this objective is not consistent; however, this is because they fix $\mathcal{G}$ to be the MEC instead of using the most up-to-date essential graph for each batch. In the supplementary material, we show that the version of the objective we work with is indeed consistent. We will drop the dependence of $R$ on $\xi'$ for readability.

Below, we provide algorithms with near-optimality guarantees for Objectives 2 and 3, while motivating a practical algorithm for Objective 1.

## 3  Related Work

Causality has been widely studied in machine learning [30, 31]. Here we focus on prior research that is most relevant to our work.

Agrawal et al. [3] and Ghassami et al. [13] give near-optimal greedy algorithms for Objectives 1 and 3 respectively. Ahmaditeshnizi, Salehkaleybar, and Kiyavash [4] present a dynamic programming algorithm for an adversarial version of Objective 3, optimizing for the worst case ground truth DAG in the MEC. However, all these algorithms only apply to *single-perturbation* interventions. Both of these works use the submodularity of the two objectives. In this paper we address the exponentially large search space that arises when designing multi-perturbation interventions, a strictly harder problem.

Much existing work in experimental design for causal DAGs is focused on identifying the graph uniquely, while minimizing some cost associated with doing experiments [9, 18, 32, 21, 23]. When the MEC is large and the number of experiments is small, identifying the entire graph will be infeasible. Instead, one must select interventions that optimize a measure of the information gained about the causal graph.

Lindgren et al. [23] show NP-hardness for selecting an intervention set of at most $m$ interventions, with minimum number of perturbations, that completely identifies the true DAG. This, however, does not directly imply a hardness result for our problem.

Gamella and Heinze-Deml [10] propose an approach to experimental design for causal structure learning based on invariant causal prediction. While our approach has guarantees for objectives relating to either the whole graph or functions of the oriented edges, their work is specific to the problem of learning the direct causes of one variable.

Acharya et al. [1] consider *testing* between two candidate causal models. However, the setting differs from ours: they assume the underlying DAG is known but allow for unobserved confounding variables.

Designing multi-perturbation interventions has been previously studied in linear cyclic networks, with a focus on parameter identification [14]. Here we focus on causal graph identification in DAGs.

## 4  Greedy Algorithms for Experiment Design

All of our algorithms follow the same general strategy. Like in previous works on single-perturbation experimental design [3, 13], we greedily add interventions to our intervention set. We add $I$ maximizing $F(\xi \cup \{I\})$ where $\xi$ is the currently proposed set of interventions. This greedy selection is justified because our objectives are submodular, a property we define formally later. For single-perturbation experimental design, this is algorithmically simple since there are only $p$ possible interventions. However, for multi-perturbation interventions even selecting greedily is *intractable* at scale since we have $\binom{p}{q}$ possible interventions. Therefore we provide ways to find an intervention

that is approximately greedy, i.e, an intervention with marginal improvement in objective that is close to that of the greedy intervention.

A further challenge with the greedy approach is that it involves evaluating the objective, which for our objectives is a potentially *exponential sum* over members of an essential graph. Each of the two algorithms we give has a different strategy for overcoming this.

In Section 4.1, we present DOUBLE GREEDY CONTINUOUS (DGC) for optimizing Objective 3, the edge-orientation objective. For greedily selecting interventions to maximize an exponential sum, we employ the stochastic continuous optimization technique of Hassani et al. [15].

In Section 4.2, we present SEPARATING SYSTEM GREEDY (SSG) for optimizing Objective 2, MI-$\infty$. To greedily select interventions, we use the construction of *separating systems* (SS) [37, 32, 23], to create a smaller set of interventions to search over. Collectively, the interventions in the SS fully orient the graph. To handle tractably evaluating the objective, we use the idea of Ghassami et al. [13] and Agrawal et al. [3] to optimize an approximation of the objective constructed using a limited sample of DAGs.

To give near-optimality guarantees for these algorithms, we will use two properties of the objectives: monotonicity and submodularity.

**Definition 1.** A set function $F : 2^V \to \mathbb{R}$ is *monotonically increasing* if for all sets $I_1 \subseteq I_2 \subseteq V$ we have $F(I_1) \leq F(I_2)$.

**Definition 2.** A set function $F : 2^V \to \mathbb{R}$ is *submodular* if for all sets $I_1 \subseteq I_2 \subseteq V$ and all $v \in V \setminus I_2$ we have $F(I_1 \cup \{v\}) - F(I_1) \geq F(I_2 \cup \{v\}) - F(I_2)$.

Submodularity is a natural diminishing returns property, and many strategies have been studied for optimizing submodular objectives [22]. In both the above definitions, $V$ is called the *groundset*, the set that we can choose elements from. In the single-perturbation problem, the groundset is just $[p]$, whereas in our case it is all subsets of up to $q$ nodes.

We show that DGC achieves an objective value within a constant factor of the optimal intervention set on Objective 3. SSG does not achieve a constant-factor guarantee, but for both infinite sample objectives we obtain a lower bound on its performance.

All of our algorithms run in polynomial time; however, they assume access to a uniform sampler across all DAGs in the essential graph. This exists for sampling from the MEC [39] but not for essential graphs given existing interventions. In practice, we find that an efficient non-uniform sampler [13] can be used to achieve strong empirical performance.

## 4.1 Optimizing the Edge-orientation Objective

In the following, we develop an algorithm for maximizing Objective 3. In fact, the algorithm we provide has a near-optimality guarantee for a more general form of $F_{\text{EO}}$, namely

$$F_{\text{EO}}(\xi) = \sum_{G \in \mathcal{G}} a(G) \sum_{e \in G} w(e) \mathbb{1}(e \in R(\xi, G)),$$

where $\forall e, w(e) \geq 0$ and $\forall G, a(G) \geq 0$. The weights $a(G)$ can be thought of as corresponding to having a non-uniform prior over the DAGs in the essential graph, whilst the weights $w(e)$ can be thought of as assigning priority to the orienting of certain edges. The inner sum above is a *weighted coverage function* [22] over the set of edges.

We will first show that $F_{\text{EO}}$ is monotone submodular over groundset $\mathcal{I}$. This generalizes a result by Ghassami et al. [13] who showed the same result for groundset $[p]$ (single perturbation interventions).

**Lemma 1.** $F_{EO}$ *is monotone submodular over the groundset* $\mathcal{I}$.

*Proof.* All proofs are presented in the supplementary material unless otherwise stated. $\square$

As mentioned, we cannot use a greedy search directly since the groundset $\mathcal{I}$ is too large. Instead, we develop an algorithm for selecting an intervention with near-maximal utility compared to the greedy choice. In particular, our strategy is to prove a submodularity result over the function $F$ with modified domain. Consider the set function $F_{\text{EO}}^{\xi}(I) = F_{\text{EO}}(\xi \cup \{I\})$ for fixed $\xi$.

**Lemma 2.** $F_{EO}^{\xi}$ *is non-monotone submodular over the groundset* $[p]$.

The Non-monotone Stochastic Continuous Greedy (NMSCG) algorithm of Mokhtari, Hassani, and Karbasi [27] can therefore be used as a subroutine to select, in expectation, an approximately greedy intervention to add to an existing intervention set. The algorithm uses a stochastic gradient-based method to optimize a continuous relaxation of our objective, and then rounds the solution to obtain a set of interventions. The continuous relaxation of $F_{EO}^{\xi}$ is the multilinear extension

$$f_{EO}^{\xi}(x) = \sum_{I \in \mathcal{I}} F_{EO}^{\xi}(I) \prod_{i \in I} x_i \prod_{i \notin I}(1 - x_i)$$

with constraints $\sum_i x_i \leq q$, $0 \leq x_i \leq 1$ for all nodes $i$. The multilinear extension can be thought of as computing the expectation of $F_{EO}^{\xi}(I)$, when input $x$ is a vector of independent probabilities such that $x_i$ is the probability of including node $i$ in the intervention. The sum over $\mathcal{I}$ in $f_{EO}^{\xi}$ and the sum over DAGs in $F_{EO}^{\xi}$ make computing the gradient of this objective intractable. Therefore, we compute an unbiased stochastic approximation of the gradient $\nabla f_{EO}^{\xi}(x)$ by uniformly sampling a DAG $G$ from $\mathcal{G}$ and intervention $I$ from the distribution specified by $x$. Define

$$\hat{f}_{EO}^{\xi}(I, G) = |R(\xi \cup \{I\}, G)|. \tag{4}$$

Mokhtari, Hassani, and Karbasi [27] show that an unbiased estimate of the gradient of $f(x)$ can be computed by sampling $G$ and $I$ to approximate

$$\frac{\partial}{\partial x_i} f_{EO}^{\xi}(x) = \mathop{\mathbb{E}}_{G, I|x} \left[ \hat{f}_{EO}^{\xi}(I, G; I_i \leftarrow 1) - \hat{f}_{EO}^{\xi}(I, G; I_i \leftarrow 0) \right], \tag{5}$$

where $I_i \leftarrow 0$ means that if $i \in I$, remove it. The use of a stochastic gradient means that $F_{EO}$ can be efficiently optimized despite it being a possibly exponential sum over $\mathcal{G}$.

After several gradient updates we obtain a vector of probabilities $x$ that approximately maximizes $f_{EO}$. To obtain an intervention $I$ one uses a ROUND function, for example pipage rounding which, on submodular functions, has the guarantee that $\mathbb{E}[F_{EO}^{\xi}(\text{ROUND}(x))] = f_{EO}^{\xi}(x)$ [6].

**Theorem 1** (Mokhtari, Hassani, and Karbasi [27]). *Let* $I^*$ *be the maximizer of* $F_{EO}^{\xi}$. NMSCG *with pipage rounding, after* $\mathcal{O}\left(p^{5/2}/\epsilon^3\right)$ *evaluations of* $R$, *achieves a solution* $I$ *such that*

$$\mathbb{E}\left[F_{EO}^{\xi}(I)\right] \geq \frac{1}{e} F_{EO}^{\xi}(I^*) - \epsilon.$$

The original result measures runtime in terms of the number of times we approximate the gradient in Equation 5 with a single sample (in our case a single $G, I$ tuple). From Equations 4 and 5 we can see that the number of gradient approximations is a constant factor of the number of evaluations of $R$. Hence, we measure runtime in terms of number of evaluations of $R$. The bottleneck for evaluating $R$ is applying the Meek Rules, which can be computed in time polynomial in $p$ [26]. Note that the NMSCG subroutine can be modified to stabilize gradient updates by the approximation of a Hessian, in which case the same guarantee can be achieved in $\mathcal{O}\left(p^{3/2}/\epsilon^2\right)$ [15]. We use this version of NMSCG for the experiments.

Our main result now follows from the fact that selecting interventions approximately greedily will lead to an approximation guarantee due to lemma 1.

**Theorem 2.** *Let* $\xi^* \in C_{N,b}$ *be the maximizer of Objective 3.* DGC *will, after* $\mathcal{O}\left(m^4 p^{5/2}/\epsilon^3\right)$ *evaluations of* $R$, *achieve a solution* $\xi$ *such that*

$$\mathbb{E}[F_{EO}(\xi)] \geq \left(1 - \frac{1}{e^{1/e}}\right) F_{EO}(\xi^*) - \epsilon.$$

We highlight this is a constant-factor guarantee with respect to the optimal batch of interventions. Our bound requires compute that is low order in $p$ with no dependence on $q$. Without even accounting for computing the possibly exponential sum in the objective, merely enumerating all possible interventions for fixed $q$ is $\mathcal{O}(p^q)$.

Ghassami et al. [13] give a similar constant-factor guarantee for batches of single-perturbation interventions ($q = 1$). Their bound is within $1 - \frac{1}{e}$ of the *optimal single-perturbation batch*. In the

**Algorithm 1** DOUBLE GREEDY CONTINU-
OUS(DGC)

---
**Input:** essential graph $\mathcal{G}$, constraints $C_{m,q}$,
objective $F_{\text{EO}}$
Init $\xi \leftarrow \emptyset$
**while** $|\xi| \leq m$ **do**
   $I \leftarrow \text{ROUND}(\text{NMSCG}(F_{\text{EO}}^{\xi}, q))$
   $\xi \leftarrow \xi \cup \{I\}$
**end while**
**Output:** a set $\xi$ of interventions

**Algorithm 2** SEPARATING SYSTEM
GREEDY(SSG)

---
**Input:** essential graph $\mathcal{G}$, constraints $C_{m,q}$,
objective $\tilde{F}_{\infty}$
Init $\mathcal{I} \leftarrow \emptyset$
$\mathcal{S} \leftarrow \text{SEPARATE}(q, \mathcal{G})$
**while** $|\xi| \leq m$ **do**
   $I' \leftarrow \arg\max_I \tilde{F}_{\infty}(\mathcal{I} \cup \{I\})$
   $\xi \leftarrow \xi \cup \{I'\}$
**end while**
**Output:** a set $\xi$ of interventions

supplementary material, we show that the optimal multi-perturbation intervention can orient up to $q$ times more edges than the optimal single perturbation intervention. In Section 5 we experimentally verify the value of multi-perturbation interventions by comparing DGC to the algorithm presented in Ghassami et al. [13]. If we allow for soft interventions, the $1 - \frac{1}{e}$ guarantee can also be obtained for multi-perturbation interventions. See the supplementary material for details.

## 4.2 Optimizing the Mutual Information Objective

We now consider an algorithm for maximizing Objective 2. First, we note that computing the sum over $\mathcal{G}$ and the size of $\xi \cup \xi'$-essential graphs is computationally intractable. The computation of $|\text{Ess}^{\xi \cup \xi'}(G)|$ makes $F_{\infty}$ a nested sum over DAGs. Like Agrawal et al. [3], we optimize a computationally tractable approximation to $F_{\infty}$. First, we uniformly sample a multiset of DAGs $\tilde{\mathcal{G}}$ from $\mathcal{G}$ to construct our approximate objective

$$\tilde{F}_{\infty}(\xi) = -\frac{1}{|\tilde{\mathcal{G}}|} \sum_{G \in \tilde{\mathcal{G}}} \log_2 |\tilde{\text{Ess}}^{\xi \cup \xi'}(G)|,$$

where $\tilde{\text{Ess}}^{\xi \cup \xi'}$ is the submultiset of $\tilde{\mathcal{G}}$ consisting of elements in $\text{Ess}^{\xi \cup \xi'}$. A submodularity result similar to lemma 1 can also be proven for $\tilde{F}_{\infty}$.

**Lemma 3.** $\tilde{F}_{\infty}$ *is monotone submodular.*

We first show that an approach similar to that used by DGC does not so easily give a near-optimal guarantee. Similarly to above, define $\tilde{F}_{\infty}^{\xi}(I) = \tilde{F}_{\infty}(\xi \cup \{I\})$ for fixed $\xi$.

**Proposition 1.** *There exists $\mathcal{G}, \tilde{\mathcal{G}}$ such that $\tilde{F}_{\infty}^{\xi}$ is not submodular.*

Hence we cannot use existing algorithms for submodular optimization to construct near-greedy interventions. We instead take a different approach. Suppose we can reduce $\mathcal{I}$ to some set of interventions $\mathcal{S}$ much smaller than $\mathcal{I}$, such that $\tilde{F}_{\infty}(\mathcal{S})$ has the maximum possible objective value. Due to lemma 3, we can obtain a guarantee by greedily selecting $m$ interventions from $\mathcal{S}$. A method for constructing the set $\mathcal{S}$ comes from *separating system* constructions.

**Definition 3.** A $q$-sparse $\mathcal{G}$-separating system of size $N$ is a set of interventions $\mathcal{S} = \{S_1, S_2 ..., S_N\}$ such that $|S_i| \leq q$ and for every undirected edge $(i, j) \in \mathcal{G}$ there is an element $S \in \mathcal{S}$ such that exactly one of $i, j$ is in $S$ [32].

A separating system of $\mathcal{G}$ completely identifies the true DAG, and hence obtains the maximum possible objective value 0 without necessarily satisfying the constraints $C_{m,q}$. As an example, in Figure 1(b,c) we see 1 and 2–sparse separating systems respectively.

We will make use of an algorithm SEPARATE$(q, \mathcal{G})$ which efficiently constructs a $q$-sparse separating system of $\mathcal{G}$. Wegener [37] and Shanmugam et al. [32] give construction methods that are agnostic to the structure of $\mathcal{G}$ (it will identify any DAG with $p$ nodes). Lindgren et al. [23] give a construction method that depends on the structure of $\mathcal{G}$.

Since the separating system obtains the maximum objective value, we can greedily select $m$ interventions from this set and obtain a lower bound on the objective value due to submodularity (lemma 3).

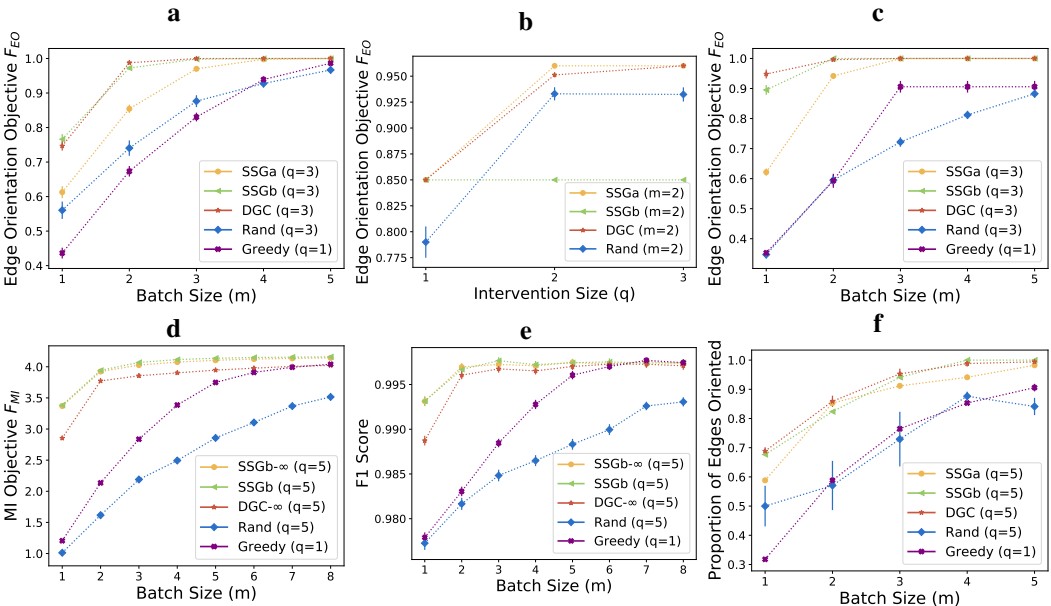

Figure 2: a–c and f give infinite sample experiments and d–e give finite samples. **a)** Our algorithms ($q = 3$) orient more edges than random interventions ($q = 3$) and greedily chosen $q = 1$ interventions for $p = 40$, ER(0.1) graphs. **b)** On a fully connected graph (p=5, m=2), SSG-B does not improve as $q$ increases. **c)** On a $p = 20$ forest of 3 disconnected star graphs, SSG-A cannot orient the full graph as quickly as our alternative approaches ($q = 3$). **d)** For finite-samples, $p = 40$, ER(0.1) graphs, the proposed methods ($q = 5$) achieve greater objective value than greedy $q = 1$ interventions. Optimizing the finite sample objective directly yields slightly greater objective value than the infinite sample approximations. **e)** The F1 scores for predicting the presence of each edge correspond well with the objective in d). **f)** For a $p = 50$ yeast subnetwork from the DREAM3 challenge, our algorithms ($q = 5$) orient more edges in the ground truth DAG than random or $q = 1$ greedy algorithms with the same batch size.

**Theorem 3.** *For $q \leq \lfloor p/2 \rfloor, m \leq |\mathcal{S}|$, SSG outputs $\xi \in C_{m,q}$ with objective value*

$$\tilde{F}_\infty(\xi) \geq (1 - \frac{m}{\lceil p/q \rceil \lceil \log p \rceil})\tilde{F}_\infty(\emptyset)$$

*in $\mathcal{O}(m|\tilde{\mathcal{G}}|\frac{p}{q} \log p)$ evaluations of $R$, when using SEPARATE as in Shanmugam et al. [32].*

Increasing $q$ does not necessarily increase $\tilde{F}_\infty(\xi)$. However, the bound we give becomes more favourable as $q$ increases because the upper bound on $|\mathcal{S}|$ decreases. In practice, we run SSG $\forall q' \leq q$ and pick the intervention set with the highest objective value on $\tilde{F}_\infty$.

Note that SSG can also be used with a similar guarantee for an analogous approximation of $F_{\text{EO}}$.

## 5 Experiments

To evaluate our algorithms we consider three settings. Firstly, randomly generated DAGs, using infinite samples per intervention. Secondly, randomly generated DAGs with linear SEMs, using finite samples per intervention. Finally, subnetworks of GRN models, using infinite samples per intervention. Full details on all of our experiments can be found in the supplementary material. For code to reproduce the experiments, see `https://github.com/ssethz/multi-perturbation-ed`.

**Infinite samples** We evaluate our algorithms using Objective 3. We consider selecting a batch of experiments where only the MEC is currently known. We vary the type of random graph and the constraint set $C_{m,q}$. The following methods are compared:

- RAND: a baseline that for $m$ interventions, independently and uniformly at random selects $q$ nodes from those adjacent to at least one undirected edge;

- GREEDY: greedily selects a single-perturbation intervention as in Ghassami et al. [13];

- DGC: our stochastic optimization approach;

- SSG-A: our greedy approach using the graph agnostic separating systems of Shanmugam et al. [32];

- SSG-B: as above, using the graph-dependent separating system constructor of Lindgren et al. [23].

Since there are infinite samples per intervention, the exact SEM used to generate data is not relevant. We plot the mean proportion of edges identified and error bars of 1 standard deviation over 100 repeats. Noise between repeats is due to randomness in the graph structure and in the algorithms themselves.

In Figure 2 a) we display the results for Erdös-Renyí random graphs with edge density 0.1 (ER(0.1)) and 40 nodes. To prevent trivial graphs and large runtimes, we only consider graphs with MEC sizes in the range $[20, 200]$. The observations given here were also found for denser Erdös-Renyí graphs and tree graphs, in addition to graphs with less nodes.

For all constraint values, all the algorithms improve greatly over RAND and GREEDY. SSG-B outperforms SSG-A, likely because the graph-sensitive separating system construction tends to return a groundset of more effective interventions.

SSG-B achieves similar objective value to DGC. However, DGC behaves most robustly when the graph structure is chosen adversarially. For example, consider Figure 2 b). Here we plot the proportion of identified edges on a $p = 5$ fully connected graph. On this graph, the separating system construction of Lindgren et al. [23] will always return the set of all single-node interventions. Therefore, its performance does not improve with $q$, whilst DGC's does. An adversarial example for SSG-A is constructed in Figure 2 c): an MEC that consists of 3 disconnected star graphs with 7, 7 and 6 nodes. In this case, DGC and SSG-B can orient most of the graph in a single intervention, whereas SSG-A will likely not contain such an intervention in the separating system it constructs.

**Finite samples**  We use linear SEMs, with weights generated uniformly in $[-1, -0.25] \cup [0.25, 1]$. Measurement noise is given by the standard normal distribution. The underlying DAGs are generated in the same way as the infinite sample experiments. Before experiment selection, we obtain 800 observational samples of the system. 3 samples are obtained for each intervention selected by our algorithms. Each perturbation fixes the value of a node to 5. We approximate Objective 1 using the methods of Agrawal et al. [3]. In particular, an initial distribution over DAGs is estimated by bootstrapping the observational data and using the techniques of Yang, Katcoff, and Uhler [40] to infer DAGs. Each DAG in the distribution is weighted proportionally to the likelihood of the observational data given the DAG and the maximum likelihood estimate of the linear model weights. The posterior over DAGs after interventions is computed by re-weighting the existing set of DAGs based on the likelihood of the combined observational and interventional data. To ensure the distribution has support near the true DAG, we include all members of the true DAG's MEC in the initial distribution.

With finite samples, our methods do not have guarantees but can be adapted into practical algorithms:

- GREEDY: greedily optimize Objective 1 with single perturbation interventions (Agrawal et al. [3]);

- DGC-$\infty$: optimizes Objective 3, with the summation over DAGs being a weighted sum over the DAGs in the initial distribution;

- SSG-B: optimizes $F_{\text{MI}}$, greedily selecting from the separating system of Lindgren et al. [23];

- SSG-B-$\infty$: approximates the objective using Objective 2 and optimizes with SSG.

For each algorithm, we record $F_{\text{MI}}$ of the selected interventions, over 200 repeats. In Figure 2 d) the objective values obtained by each algorithm are shown for varying batch size and $q = 5$. DGC-$\infty$ performs worse than SSG-B and SSG-B-$\infty$, perhaps because it is optimizing $F_{\text{EO}}$ which is not totally aligned with $F_{\text{MI}}$. Between SSG-B and SSG-B-$\infty$, there was a small benefit to directly optimizing $F_{\text{MI}}$ as opposed to its infinite sample approximation $F_{\infty}$. Accounting for finite samples may lead to greater improvements when there is heteroscedastic noise or a wider range of weights.

Performance on $F_{\text{MI}}$ corresponds closely with performance on a downstream task as shown in Figure 2 e). For each algorithm, we compute the posterior over DAGs given the selected interventions. Then, we independently predict the presence of each edge in the true DAG. Figure 2 e) plots the average F1 score for each algorithm. In finite samples, our approaches outperform both RAND and GREEDY.

**DREAM 3 networks**    We evaluate our algorithms under infinite samples using subgraphs of GRN models. From the "DREAM 3 In Silico Network" challenge [24], we use the 5 subgraphs with $p = 50$ nodes. Here, we present the results for "Yeast1", which is the graph requiring the most interventions for our methods to orient. Results for the other graphs are in the supplementary material.

We compare the same algorithms as considered in the other infinite sample experiments. In Figure 2 f) we record the proportion of unknown edges that were oriented by each algorithm. Our methods and RAND all have intervention sizes of 5. For each method we perform 5 repeats. On Yeast1, our methods all perform similarly and outperform RAND and GREEDY. This is found for the other DREAM 3 graphs too, except on one subgraph where GREEDY performs similarly to our methods.

## 6    Conclusions

We presented near-optimal algorithms for causal structure learning through multi-perturbation interventions. Our results make novel use of submodularity properties and separating systems to search over a doubly exponential domain. Empirically, we demonstrated that these algorithms yield significant improvements over random interventions and state-of-the-art single-perturbation algorithms. These methods are particularly relevant in genomics applications, where causal graphs are large but multiple genes can be intervened upon simultaneously.

## Acknowledgments and Disclosure of Funding

This research was supported in part by the Swiss National Science Foundation, under NCCR Automation, grant agreement 51NF40 180545. Caroline Uhler was partially supported by NSF (DMS-1651995), ONR (N00014-17-1-2147 and N00014-18-1-2765), IBM, and a Simons Investigator Award.

Thank you to Raj Agrawal for a helpful discussion regarding experiments.

Experiments were performed on the Leonhard cluster managed by the HPC team at ETH Zürich

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
