# Appendix

## A.1 Potential Negative Societal Impacts

We propose algorithms for experimental design when learning causal structures. The most obvious application is in designing scientific experiments to learn gene regulatory networks. It is possible that the ability to gather increasingly detailed information about gene networks could be used by malicious actors. However, to the authors' knowledge there are no examples of such use with existing related methods. On the other hand, there is longstanding scientific interest in learning such networks and potentially beneficial applications in drug design [19].

## A.2 The Power of Multi-Perturbation Interventions

We can construct a more general example than the one given in Figure 1, to demonstrate that the optimal $q$-node intervention can learn up to $q$-times more edges than the optimal single node intervention. Consider a graph with MEC that is a forest of undirected star graphs, each with an equal number of nodes. The optimal single-node intervention can intervene on the center node in one of these stars, and entirely orient that star. The optimal $q$-node intervention can achieve this for $q$ of the stars.

This example illustrates the upper bound on the number of additional edges that can be oriented by the optimal $q$-node intervention compared to the optimal single node intervention. It can be seen that this is a tight upper bound from the submodularity of $F_{\text{EO}}^{\xi}$ in lemma 2.

## A.3 Deriving $F_\infty$

Agrawal et al. [3] derive this infinite sample objective for the case of having gathered only infinite observational data before doing experiments. For the case of having infinite samples from observational data and some interventions, we follow a similar argument.

For evaluating $U_{M.I}(y, \xi, D) = H(G \mid D) - H(G \mid D, y, \xi)$ where $H$ is the entropy, we only consider the second term. The first term does not depend on the interventions we do so is irrelevant for optimizing the objective.

Given $D$ consisting of existing interventions $\xi'$ and observational data, the true DAG is already recovered up to it's $\xi'$-MEC. After obtaining infinite samples from each intervention in intervention set $\xi$, we recover the true DAG up to its $\xi' \cup \xi$-MEC. Therefore $H(G \mid D, y, \xi) = \log_2(|Ess^{\xi \cup \xi'}(G)|)$ when the true DAG is $G$. $Ess^{\xi \cup \xi'}(G)$ is the essential graph obtained after interventions $\xi$. Our prior distribution over DAGs is uniform over the $\xi'$-MEC of the true DAG. Averaging over these possibilities, the final objective is given by Objective 2.

## A.4 Background on the Meek Rules

After performing interventions, the Meek rules can be used to orient additional edges. The rules are given in Figure 3. The Meek rules are continually applied until none of the left side patterns appear in the partially directed graph.

## A.5 $F_{\text{EO}}$ is consistent

We use the same definition of *budgeted batch consistency* introduced in Agrawal et al. [3].

**Definition 4** (Agrawal et al. [3])**.** Assume our goal is to identify the true DAG $G^*$. Let us have constraints for the $b$th batch ($0 \leq b \leq B$) of experiments $C_{m,q}^b$ ($q \geq 1$). Objective $F$ is budgeted batch consistent if maximizing it in every batch implies

$$\mathbb{P}(G \mid D_B) \to \mathbb{1}(G = G^*)$$

asymptotically as $m, B \to \infty$, where $D_B$ is the combined data obtained from all batches of experiments and the original dataset.

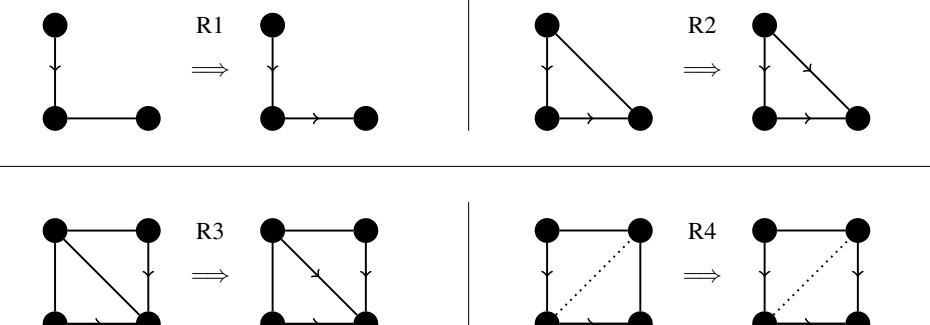

Figure 3: The 4 Meek rules. When a pattern on the left of the implication occurs, edges are oriented according to the pattern on the right of the implication. A dashed line means the direction of the edge may or may not be already identified. The name of each Meek rule is given above the implication sign. R0 refers to the step of orienting edges before Meek rules are applied.

We show that our definition of $F_{\text{EO}}$ satisfies budgeted batch consistency. Then we explain how the slight difference in the definition of $F_{\text{EO}}$ given in Agrawal et al. [3] leads to the authors concluding that it is not budgeted batch consistent.

**Proposition 2.** *$F_{EO}$ is budgeted batch consistent.*

*Proof.* Since each intervention offers an infinite number of samples, we can reason directly about the subsequent orienting of edges due to the obtained samples. Let the set of interventions after $b$ batches be $\xi_b$. We simply need to prove that as $k \to \infty$, $\xi_b$ will identify the orientation of every edge in the true DAG. Equivalently, we show that if $\xi_b$ has not oriented every edge (the current essential graph has size greater than 1), $\xi_{b+1}$ will orient additional edges in the true DAG. There are only a finite number of edges to be oriented, so this means $\xi_\infty$ will fully identify the true DAG.

If $\xi_b$ does not orient every edge in the true DAG, Objective 3 will have a maximum in the next batch of greater than 0. Moreover, to obtain objective greater than 0, the interventions selected in batch $b + 1$ must orient edges not oriented by $\xi_b$. This is because after each batch, the objective is updated to be with reference to the essential graph of $G$ under interventions in $\xi_b$. To orient unidentified edges in any of the possible DAGs (those in the current $\xi_b$-MEC), we must orient at least one edge in the true DAG after obtaining samples. This can be seen from the first step of computing function $R$. One can also see that if an edge can be oriented, it can always be oriented by selecting a single unique intervention of size 1 and thus the constraints $C_{m,q}$ can always be satisfied. $\square$

The key difference between this argument and the one given by Agrawal et al. [3] is that they maintain a static objective function between batches, so the same set of interventions is selected every round.

### A.6 Proof of Lemma 1

We'll work with the notation of $\xi_1 \subset \xi_2$ are sets of interventions, and $I$ is an intervention. We'll write $R(\xi, G) = M(A(\xi, G), G)$. Here, $A$ carries out the first step of orienting edges based on one of the nodes in that edge being intervened on (we refer to this as R0). $M$ carries out the Meek rules given the orientations of edges in $A$. Note that $R, M, A$ implicitly depend on the interventions carried out in previous batches $\xi'$, since this determines what edges might already be oriented in $G$ (whether we are orienting the MEC or some essential graph).

We'll first show that function $R$ has a monotonicity-like property: adding an intervention only adds to the set of oriented edges.

**Proposition 3.** *Monotonicity-like property of R: $R(\xi_1, G) \subseteq R(\xi_2, G)$ for all $G$, $\xi_1 \subseteq \xi_2$.*

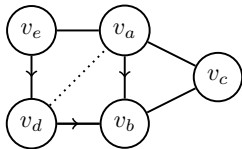

Figure 4: The pattern when an edge $v_a \to v_b$ is identified by R4 and $v_a - v_c$, $v_b - v_c$

.

*Proof.* The same argument is given in Ghassami et al. [13]. By the definition of $A$, $A(\xi_1, G) \subseteq A(\xi_2, G)$. The Meek rules are sound and order-independent [26], and therefore $M(A(\xi_1, G)) \subseteq M(A(\xi_2, G))$ □

From this we can see that $F_{\text{EO}}$ is also monotonic.

To prove lemma 1, we swill first prove some propositions regarding the marginal change in $R$ when adding a new intervention.

**Proposition 4.** *Consider vertices $v_a$, $v_b$, $v_c$. Consider $\bar{G}$, the partially directed graph obtained after doing intervention set $\xi$ and then applying the Meek rules exhaustively. If $v_a \to v_b \in \bar{G}$, and $v_b - v_c \in \bar{G}$, then $v_a \to v_c \in \bar{G}$.*

*Proof.* If $v_a \to v_b$, and $v_b - v_c$, we must have that $v_a$ and $v_c$ are adjacent, else $v_b \to v_c$ by R1. We cannot have $v_c \to v_a$ since this would identify $v_b - v_c$ by R2. Hence we have either that $v_a - v_c$ or $v_a \to v_c$.

Suppose for contradiction that, after applying all Meek rules, for some nodes $v_a, v_b, v_c$ we have $v_a \to v_b$, $v_b - v_c$ and $v_a - v_c$. We will gain a contradiction by an infinite descent. Any DAG can be associated with some permutation of its nodes that specifies a topological ordering, with the lowest ranked node being the root. Suppose that $B$ is the lowest ranked node in the topological ordering given by the true DAG (closest to the root) such that the supposed pattern holds. For all ways in which $v_a \to v_b$ could have been identified, we will show that either in fact $v_a \to v_c$ or find a vertex lower than $v_b$ in the topological ordering that fits into an identical pattern.

Some cases are covered in Meek [26] when proving a similar result (lemma 1).

Suppose $v_a \to v_b$ is known by being a collider (identified before any interventions take place). This is handled in Meek [26].

Suppose $v_a \to v_b$ is known by R1, R2, R3. These cases are all handled by Meek [26].

Suppose $v_a \to v_b$ is learnt directly by an intervention (rule R0). If it is identfied in this way, there must exist an intervention $I$ such that exactly one of $v_a, v_b \in I$. However, in either case regardless of whether $v_c \in I$ or not, we identify one of $v_a - v_c$ or $v_b - v_c$ by R0. Hence the pattern cannot occur if $v_a \to v_b$ is identified by R0.

Suppose $v_a \to v_b$ is learnt directly by R4 as in Figure 4. Consider the pattern given in Figure 4 where $v_e$ and $v_d$ are the other edges in the R4 pattern. Now if $v_a - v_d$ is undirected, then $AED$ is the same pattern as $v_a, v_b, v_c$ but with $v_d < v_b$ in the topological ordering (a contradiction). If $v_a \to v_d$, then $v_a, v_d, v_b$ gives the same setup as if we discovered $v_a \to v_b$ through R2. Similarly, if $v_d \to v_a$, then $v_e \to v_a$ by R2 and then we have the same setup as if we discovered $v_a \to v_b$ by R1. An alternative R4 pattern can also orient $v_a \to v_b$, however it involves node $v_c$ being part of the pattern. In this case, we must have oriented $v_c \to v_b$, a contradiction. □

Proposition 4 allows us to prove two propositions more directly related to our final result.

**Proposition 5.** $R(\xi_2 \cup \{I\}) \setminus R(\xi_2) \subseteq R(\xi_1 \cup \{I\}) \setminus R(\xi_1)$

*Proof.* Here we'll take $R$ to also include all edges oriented before the interventions, since this doesn't change the outcome of the set difference operation above. We also drop $G$ from the notation since we work with a fixed true graph. This is just done out of convenience for the proof.

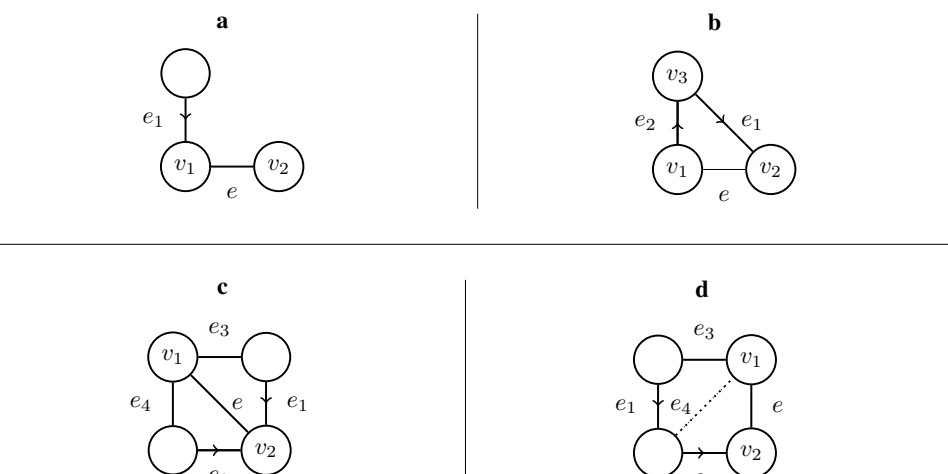

Figure 5: The Meek rules with labels on edges and nodes. **a)** R1, **b)** R2, **c)** R3, **d)** R4.

Take edge $e \in R(\xi_2 \cup \{I\})$ and $e \notin R(\xi_2)$. By the monotonicity-like property, $R(\xi_1) \subseteq R(\xi_2)$, so $e \notin R(\xi_1)$. Thus we just need to prove that for all such $e$, we have $e \in R(\xi_1 \cup \{I\})$.

Assume for contradiction there is some nonempty set $E^\dagger$ of edges such that $\forall e \in E^\dagger$, $e \in R(\xi_2 \cup \{I\})$ and $e \notin R(\xi_2)$, but $e \notin R(\xi_1 \cup \{I\})$. We can specify an ordering over these edges. Order the edges (written $v_1 \to v_2$) such that: the edges are in increasing order upon the position of $v_2$ in the topological ordering of graph $G$ (lower is closer to the root). Settle all ties by decreasing order on the position of $v_1$ in the topological ordering. We will now show that if there exists some $e$ that is the lowest ordered element in $E^\dagger$, we can always either create a contradiction or find some alternative edge in $E^\dagger$ with lower position in the ordering (itself a contradiction).

If $e = v_1 \to v_2$ is discovered in $R(\xi_2 \cup \{I\})$ by R0, we know $I$ must intervene on one of $v_1$, or $v_2$ but not the other. Hence $e \in R(\xi_1 \cup \{I\})$.

The diagram in Figure A.6 is given for following the other cases. Notation refers to the names of nodes and edges given on the diagrams.

If $e = v_1 \to v_2$ is discovered in $R(\xi_2 \cup \{I\})$ by R1, we must have $e_1 \in R(\xi_2 \cup \{I\})$. We must also have that $e_1 \notin R(\xi_2)$ else $e \in R(\xi_2)$ by R1. The same must be true of $e'$ not being in $R(\xi_1 \cup \{I\})$, so $e_1 \in E^\dagger$. But $e_1$ will have lower ordering since $v_1$ is below $v_2$ in the topological ordering.

If $e = v_1 \to v_2$ is discovered in $R(\xi_2 \cup \{I\})$ by R2, we must have some pair $e_1, e_2 \in R(\xi_2 \cup \{I\})$ such that $e_1 = v_3 \to v_2$ and $e_2 = v_1 \to v_3$. One of these edges cannot be in $R(\xi_2)$. In fact, neither can be in $R(\xi_2)$ due to proposition 4. This is because assuming only one is identified, based on proposition 4, another edge is either identified which leads to orienting $e$ or another edge is incorrectly oriented in the true DAG. The same holds for $R(\xi_1 \cup \{I\})$, but then we've found an edge $e_2 \in E^\dagger$ that is lower in the ordering than $e$.

If $e = v_1 \to v_2$ is discovered in $R(\xi_2 \cup \{I\})$ by R3, we must have $e_1, e_2 \in R(\xi_2 \cup \{I\})$ and $e_3, e_4 \in G$ but not necessarily in $R(\xi_2 \cup \{I\})$. $e_1$ and $e_2$ form an unshielded collider and are identified before intervening, so $e \in R(\xi_1 \cup \{I\})$ by R3.

If $e = v_1 \to v_2$ is discovered in $R(\xi_2 \cup \{I\})$ by R4, we must have $e_1, e_2 \in R(\xi_2 \cup \{I\})$ and $e_3, e_4 \in G$ but not necessarily in $R(\xi_2 \cup \{I\})$. At least one of $e_1, e_2 \notin R(\xi_2)$. Suppose only $e_1$ is in, then $e_2$ is in by R1. Therefore $e_1$ is not in $R(\xi_2)$ or $R(\xi_1 \cup \{I\})$, but this is lower in the ordering than $e$. $\qquad\square$

**Proposition 6.** $R(\xi_1) \setminus R(\xi_1 \cup \{I\}) \subseteq R(\xi_2) \setminus R(\xi_2 \cup \{I\})$

*Proof.* Follows by monotonicity of $R$, both sides are the empty set. $\qquad\square$

We can rewrite $F_{EO} = \sum_{G \in \mathcal{G}} g(R(\xi, G))$ where $g$ is the weighted coverage function. $g$ is a monotonic function of the set of edges in $R$.

$$g(R(\xi_1 \cup \{I\})) - g(R(\xi_1)) \overset{\text{(i)}}{=} g(R(\xi_1 \cup \{I\}) \setminus R(\xi_1))$$
$$- g(R(\xi_1) \setminus R(\xi_1 \cup \{I\}))$$
$$\overset{\text{(ii)}}{\geq} g(R(\xi_2 \cup \{I\}) \setminus R(\xi_2))$$
$$- g(R(\xi_2) \setminus R(\xi_2 \cup \{I\}))$$
$$= g(R(\xi_2 \cup \{I\})) - g(R(\xi_2)).$$

Step (i) is a property of the weighted coverage function. Step (ii) comes from propositions 5 and 6 and the monotonicity of $g$. This shows that definition 2 holds for $g(R(\xi, G))$ as a function of $\xi$ for all $G$. Since the sum of submodular functions is submodular, this implies lemma 1.

### A.7 Proof of Lemma 2

We can see that a monotonicity property like proposition 3 does not hold in this case. The intervention $[p]$, for example, orients no edges. Nevertheless, we follow the same approach to prove submodularity of $F_{\text{EO}}^{\xi}$. For this we consider $I_1 \subset I_2$ and consider adding node $v \notin I_2$ to these interventions.

For notational simplicity, $R(I, G)$ will now denote all edges oriented after intervention set $\xi$ (fixed) and intervention $I$ on true graph $G$. We will drop $G$ from the notation in cases where the graph is fixed.

**Proposition 7.** $R(I_2 \cup \{v\}) \setminus R(I_2) \subseteq R(I_1 \cup \{v\}) \setminus R(I_1)$

*Proof.* We want to show two things. First we want to show that if $e \in R(I_2 \cup \{v\})$ and $e \notin R(I_2)$, then $e \in R(I_1 \cup \{v\})$. This is shown with an identical technique to the one in proposition 5. This is because the Meek rules are the same in both cases. The only difference is the case when $e = v_1 \to v_2$ is discovered by R0. In this case, $v$ must be either $v_1$ or $v_2$ and neither of $v_1$ or $v_2$ can be in $I_2$. However therefore neither are in $I_1$ and hence $e \in R(I_1 \cup \{v\})$.

The second thing we need to show is that if $e \in R(I_2 \cup \{v\})$ and $e \notin R(I_2)$, then $e \notin R(I_1)$. Suppose for contradiction that there exists some such $e \in R(I_1)$. We'll proceed in two steps. First we'll show that in order to avoid a contradiction, we must have that $I_2$ intervenes on both vertices in $e$. Second we'll show that if we intervene on both vertices in $e$ for $I_2$, we cannot have that $e \in R(I_2 \cup \{v\})$ and $e \notin R(I_2)$.

We can represent the identification of edge $e$ in $R(I_1)$ by a directed tree diagram. The root in the diagram is $e$, and the children of each node in the diagram are the directed edges involved in the Meek rule that identifies the parent edge. Each node can have either one or two children (since each Meek Rule depends on up to two specific edges being directed). Leaf nodes must have been identified by R0. Since $e \in R(I_1)$ and $e \notin R(I_2)$, there must be leaf nodes in the diagram that are not identified by R0 using intervention $I_2$. Since $I_1 \subset I_2$, this means that for these leaf edges, $I_2$ must contain both nodes. However we now show that, for the structure of all Meek rules, in our tree diagram if a child edge has both vertices intervened on then the parent is identified anyway, unless both vertices of the parent are intervened on. This is shown pictorially in Figure 6. R4 case 1 requires some extra explanation. $e_1$ is identified and must be oriented towards the node in $e$ to prevent identifying $e$ by R2. Hence $e_2$ must direct into $e$ to prevent a cycle. Then $e$ is learnt by R1. Given all this, we can see that to prevent identification of $e$ by $I_2$, there must be a path from a leaf edge to the root $e$ where all the edges in the path have both vertices in the edge intervened on in $I_2$. Hence $e$ has both vertices contained in $I_2$.

Now consider a different tree diagram for how $e$ is oriented in $R(I_2 \cup \{v\})$. Clearly $v$ must be included as a vertex in at least one of the leaf edges, else the same pattern could allow us to orient $e$ using $I_2$. We show that if both vertices of an edge are intervened on, to prevent the identification of both child edges and hence the edge itself by the Meek rules, it must be that both children have both of their vertices intervened on in $I_2$. This is shown in Figure 7.

In Figure 7, **R1**, **R2** and **R3** are clear. For **R4 case 1**, $e_2$ must orient right to left (or get $e$ by R1). Also, $e_1$ must go towards the top right or we get $e$ by R2. Thus $e_3$ must point upwards by R2 which

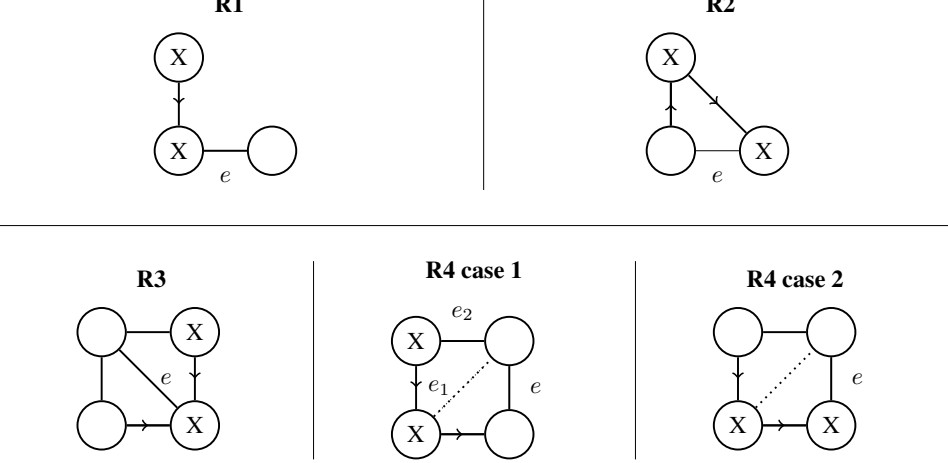

Figure 6: The pattern of each Meek rule as in the first part of proposition 7. An $X$ denotes that an intervention occured at that node. In our tree representation, if a child edge has both vertices intervened on then the parent is identified anyway, unless both vertices of the parent are intervened on.

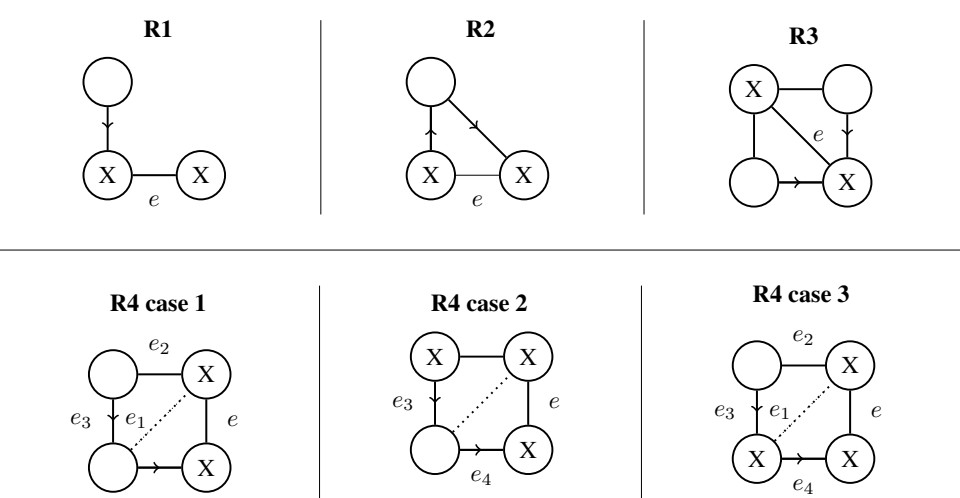

Figure 7: The pattern of each Meek rule as in the second part of the proof of proposition 7. An $X$ denotes that an intervention occured at that node. In our tree diagrams, if both vertices of an edge are intervened on, to prevent the identification of both child edges (and hence the edge itself by the corresponding Meek rule), it must be that both children have both of their vertices intervened on.

is a contradiction, since we know in the R4 pattern $e_3$ points downwards. Intervening on more edges to avoid this leads us to cases 2 and 3. For **R4 case 2**, we learn $e_3$ and $e_4$ and hence orient $e$. For **R4 case 3**, $e_2$ must point left else we get $e$ by R1. We know $e_3$ points down and it is oriented by R0. Then, $e_4$ goes left to right by R1. $E_1$ then points towards the bottom left by $R_2$ and hence $e$ is oriented by R2.

Hence, there must be some path from the root $e$ to a leaf edge such that all members of the path have both of their vertices intervened on in $I_2$ and hence $I_2 \cup \{v\}$. If $v \in e$, we have a contradiction since $v \notin I_2$. If $v \notin e$, then the leaf edge in our tree representation containing $v$ is not identified using R0

with $I_2 + v$ either. Hence, this tree cannot possibly represent the sequence of Meek rules that lead to orienting $e$. Thus if $e \in R(I_2 \cup \{v\})$ and $e \notin R(I_2)$, then $e \notin R(I_1)$. □

**Proposition 8.** $R(I_1) \setminus R(I_1 \cup \{v\}) \subseteq R(I_2) \setminus R(I_2 \cup \{v\})$

*Proof.* As in the proof of lemma 1, we can write $R(I) = M(A(I))$, where $A$ returns edges oriented directly by the intervention and $M$ returns these in addition to any oriented due to Meek rules.

By symmetry in the definition of $A$, we can see that $A(I^C) = A(I)$ and hence $R(I^C) = R(I)$. Take some edge $e \in R(I_1) \setminus R(I_1 \cup \{v\})$. Then by symmetry we have $e \in R(I_1^C) \setminus R(I_1^C \setminus \{v\})$. Then since $I_1^C \setminus \{v\} \supseteq I_2 - \{v\}$, by proposition 7 we must have $e \in R(I_2^C) \setminus R(I_2^C \setminus \{v\})$. Again by symmetry we then have $e \in R(I_2) \setminus R(I_2 \cup \{v\})$. □

We can conclude that $F_{\text{EO}}^\xi$ is submodular in an identical way to how we did in proving lemma 1, by combining propositions 7 and 8.

### A.8 Proof of Theorem 2

At each iteration of selecting an intervention, Theorem 1 lower bounds how close the marginal gain compared to the greedy intervention is. Let $\xi$ be the set of interventions DGC selects. Let $\xi^*$ be the optimal batch of interventions, and $I^*_i$ be the ith intervention in this set. Due to lemma 1 (monotonicity), $\xi^*$ contains exactly $m$ interventions. $\{\xi_i\}_{i \geq 0}$ is the entire intervention set after each greedy selection. Define marginal improvement $\Delta(I|\xi) = F_{\text{EO}}(I \cup \xi) - F_{\text{EO}}(\xi)$. The following holds for all $i$:

$$
\begin{aligned}
F_{\text{EO}}(\xi^*) &\leq F_{\text{EO}}(\xi^* \cup \xi_i) \\
&\leq F_{\text{EO}}(\xi_i) + \sum_{j=1}^{k} \Delta(I^*_j \mid \xi_i \cup \{I^*_1, ..., I^*_{j-1}\}) \\
&\leq F_{\text{EO}}(\xi_i) + \sum_{I \in \xi^*} \Delta(I \mid \xi_i) \\
&\leq F_{\text{EO}}(\xi_i) + \sum_{I \in \xi^*} e\, \mathbb{E}\left[ F_{\text{EO}}(\xi_{i+1}) - F_{\text{EO}}(\xi_i) \right] + e\epsilon \\
&\leq F_{\text{EO}}(\xi_i) + em\, \mathbb{E}\left[ F_{\text{EO}}(\xi_{i+1}) - F_{\text{EO}}(\xi_i) \right] + em\epsilon
\end{aligned}
$$

The first line is due to monotonicity. The second is a telescoping sum. The third is due to submodularity (lemma 1). The fourth is a result of Theorem 1, since what we write is lower bounded by the greedy choice, which by definition has greater marginal improvement than any other intervention. The expectation here is over noise in selecting the $(i+1)$th intervention. The final line just notes that there are $m$ elements in the sum.

Now define $\delta_i = F_{\text{EO}}(\xi^*) - F_{\text{EO}}(\xi_i)$. We rearrange the above to get

$$
\delta_i \leq em(\delta_i - \mathbb{E}[\delta_{i+1}] + \epsilon),
$$

where again the expectation is over selection of the $(i+1)$th intervention.

Now we telescope this inequality, subbing in $i + 1 = m$ to obtain our final result.

$$
\begin{aligned}
\mathbb{E}[\delta_m] &\leq \left(1 - \frac{1}{em}\right) \delta_{m-1} + \epsilon \\
&\leq \left(1 - \frac{1}{em}\right)^m \delta_0 + \sum_{j=0}^{m} \left(1 - \frac{1}{em}\right)^j \epsilon \\
&\leq e^{-1/e} \delta_0 + \sum_{j=0}^{m} e^{-\frac{j}{em}} \epsilon
\end{aligned}
$$

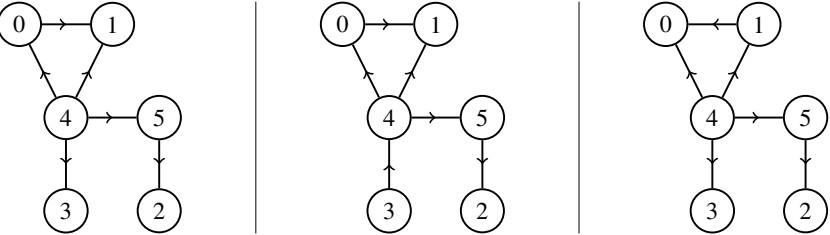

Figure 8: The 3 DAGS in $\tilde{\mathcal{G}}$ for the counterexample in proving Proposition 1.

The third line uses $1 - x \leq e^{-x}$. The final term on the right hand side of the last line reduces to $\sigma\epsilon$, where

$$\sigma = \frac{e^{-1/e} - e^{1/(em)}}{1 - e^{1/(em)}}.$$

By noting that $\delta_0 = F_{\text{EO}}(\xi^*)$, we get

$$\mathbb{E}[F_{\text{EO}}(\xi)] \geq \left(1 - e^{-\frac{1}{e}}\right) F_{\text{EO}}(\xi^*) - \epsilon\sigma.$$

Given Theorem 1 and that we select $m$ greedy interventions, this requires $\mathcal{O}\left(mp^{5/2}/\epsilon^3\right)$ calls to $R$. We remove the dependence on $m$ in $\sigma$ into the runtime by noting that $\sigma = \mathcal{O}(m)$ and hence that if we do $\mathcal{O}\left(m^4 p^{5/2}/\epsilon^3\right)$ calls to the Meek rules, we get

$$\mathbb{E}[F_{\text{EO}}(\xi)] \geq \left(1 - e^{-\frac{1}{e}}\right) F_{\text{EO}}(\xi^*) - \epsilon.$$

### A.9 Proof of Lemma 3

Agrawal et al. [3] prove monotone submodularity of $F_{\text{MI}}$. $\tilde{F}_\infty$ is the special case of this objective in the limit of infinite samples per intervention.

### A.10 Proof of Proposition 1

To show that $\tilde{F}_\infty^\xi$ is in general not submodular, we need to give a specific example of a constraint set, and set of DAGs $\tilde{\mathcal{G}}$. We let $m = 1, k = 4$. The set $\tilde{\mathcal{G}}$ is given in Figure 8. We consider the set of interventions before carrying out experiments, $\xi'$ to be the empty set. To break the definition of submodularity as given in definition 2 we need to define an existing intervention to add, $I_2$, and a subset of this, $I_1$. For this example we choose $I_2 = [1, 2, 3]$ with nodes numbered as in Figure 8. $I_1 = [1, 2]$. We also need to choose a perturbation to add to each intervention, and we choose node 0.

The computation of the objective is carried out in *proposition.py* in the accompanying code.

Note that for the special case $\tilde{\mathcal{G}} = \mathcal{G}$, we have not constructed a counterexample. In fact, in this case we suspect that $\tilde{F}_\infty^\xi$ is submodular, but don't have a proof. If true, this may suggest that an algorithm similar to DGC could be used to maximize Objective 2 directly without approximating the MEC with a bag of DAGs. An additional difficulty for this objective, however, is the second potentially exponential sum required to compute essential graph sizes embedded within the logarithm.

### A.11 Proof of Theorem 3

We know that $\tilde{F}_\infty(\mathcal{S}) = 0$, because the graph is then fully identified, meaning $|\tilde{\text{Ess}}^{\xi \cup \xi'}(G)|$ is 1 for all $G$. We also know that $\min(\tilde{F}_\infty) = \tilde{F}_\infty(\emptyset)$. The submodularity of $\tilde{F}_\infty$ over groundset $\mathcal{S} \subset \mathcal{I}$

(lemma 3) along with this boundedness of the function is used to get the final bound.

Say we greedily select $m$ members of $\mathcal{S}$ to construct $\xi_m$. We prove by induction that $Q(m) = \left( \tilde{F}_\infty(\xi_m) \geq \left( 1 - \frac{m}{|\mathcal{S}|} \right) \tilde{F}_\infty(\emptyset) \right)$ is true for all $m$ where $0 \leq m \leq |S|$.

The base case $Q(0)$ is trivial. Now assume that $Q(m)$ is true. Since $\tilde{F}_\infty$ is submodular over groundset $\mathcal{S}$, it satisfies the diminishing returns property of definition 2. Therefore it must be the case that $\exists I \in \mathcal{S} \setminus \xi_m$ such that $\tilde{F}_\infty(\{I\} \cup \xi_m) \geq \frac{1}{|\mathcal{S}|-m} \tilde{F}_\infty(\xi_m)$. Note that if this was not the case, because of submodularity, adding all of the remaining interventions in $S$ in sequence would give $\tilde{F}_\infty(\mathcal{S}) < 0$ which would be a contradiction. Therefore

$$
\begin{aligned}
\tilde{F}_\infty(\xi_{m+1}) &\geq \frac{1}{|\mathcal{S}| - m} \tilde{F}_\infty(\xi_m) \\
&\geq \frac{1}{|\mathcal{S}| - m} \left( 1 - \frac{m}{|\mathcal{S}|} \right) \tilde{F}_\infty(\emptyset) \\
&= \left( 1 - \frac{m+1}{|\mathcal{S}|} \right) \tilde{F}_\infty(\emptyset)
\end{aligned}
$$

which completes the induction. The final result follows by applying the lower-bound on $|\mathcal{S}|$ given in Shanmugam et al. [32]. A bound for the graph-sensitive separating system in Lindgren et al. [23] can also be obtained by plugging in their lower-bound on $|\mathcal{S}|$.

The runtime can be seen by observing that for each intervention, we compare $|\mathcal{S}| = \mathcal{O}(\frac{p}{q} \log p)$ possible interventions. For each we evaluate $R$ a total of $\mathcal{O}(|\tilde{\mathcal{G}}|)$ times to compute $\tilde{F}_\infty$. Computing $R$ for each $G \in \mathcal{G}$ is sufficient to compute $\tilde{F}_\infty$ because $R$ outputs all of the oriented edges in a graph given an intervention and hence can determine if a graph is in a certain interventional MEC. Thus, the overall runtime is $\mathcal{O}(m|\tilde{\mathcal{G}}|\frac{p}{q} \log p)$ evaluations of $R$. The construction of the separating system itself is efficient compared to the computation of the Meek rules required to evaluate $R$ [32].

## A.12 Extension to Soft Interventions

Our results can also be used to develop algorithms for the soft intervention setting. A hard intervention makes the value of a variable independent of its parents. However, a soft intervention modifies a variable's value whilst maintaining the dependence on its parents. A simple example of a soft intervention is adding a constant value to the intervened node. In a GRN, a soft intervention might correspond to a gene knockdown, where a gene's expression is reduced but not set to $0$.

**Definition 5** (Soft intervention). *A soft intervention $I$ on nodes $X_T$ for all $i \in I$, adds the intervention variables $W_i$ as an extra direct cause of $X_i$.*

The soft intervention setting is greatly simplified by the following result in Ghassami, Salehkaleybar, and Kiyavash [12].

**Lemma 4** (Ghassami, Salehkaleybar, and Kiyavash [12]). *In the infinite sample setting, a set of $k$ soft interventions of size $1$ is equivalent to targeting the same $k$ nodes in a single soft intervention. By equivalent, we mean it identifies the same information regarding the true DAG.*

Therefore, the problem of selecting up to $m$ interventions with size at most $q$ is equivalent to selecting $mq$ single node interventions in the hard intervention case. In Ghassami et al. [13], the authors show that for this setting a greedy algorithm can achieve a constant factor guarantee for the objective $F_{\text{EO}}$, where the constant factor will be $1 - \frac{1}{e}$ instead of $1 - \frac{1}{e^{1/e}}$. An almost identical analysis to that done in Theorem 2 can show that this guarantee is also achieved for optimizing $\tilde{F}_\infty$.

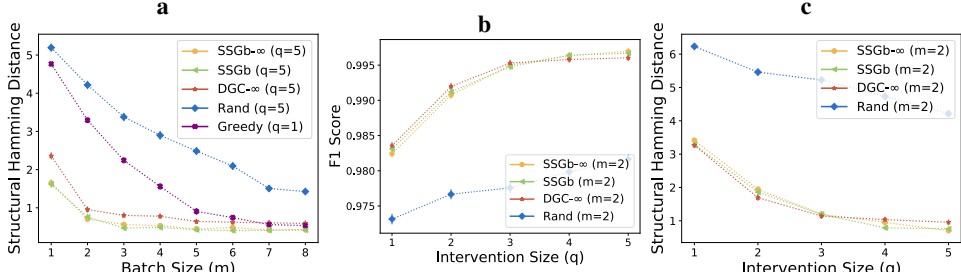

Figure 9: **a)** We measure performance on the finite sample task (same as in Figure 2(e)) with fixed intervention size ($q = 5$) using the SHD metric. The rank ordered performance of methods matches the outcome when we used the F1 metric. **b**, **c)** Here we consider the same experiment as (a) but plot F1 and SHD score respectively for changing intervention size with fixed batch size ($m = 2$). In both cases our algorithms clearly outperform a random approach.

### A.13 Further Experiment Details

All algorithms related to SSG make use of lazy evaluation for speed-up, as in Ghassami et al. [13]. In greedy selection, lazy evaluation skips interventions which, based on previous evaluations and submodularity of the objective, could not be the greedy option. This results in no change in the selected interventions but reduces the number of necessary comparisons.

When constructing graph agnostic separating systems according to the method of Shanmugam et al. [32], we compute them exactly as specified since the algorithm is fast. For the graph sensitive construction method of Lindgren et al. [23], we use approximate algorithms for constructing a minimal vertex covering and a minimal graph-coloring, since both subroutines are NP-hard [20]. For graph-coloring, we use the Welsh-Powell algorithm which is near-optimal for graphs of bounded degree [38]. For vertex cover, we use a 2-approximation algorithm based on greedily finding a maximal matching [29].

For infinite sample experiments, when approximating the objective for use in our algorithms we use a multiset of 40 DAGs uniformly sampled from the MEC with replacement. However, for evaluation we use all DAGs in the MEC.

For DGC, when using Pipage rounding, we round 10 times and select the intervention with greatest approximate objective value.

For finite-sample experiments, the approximate prior over DAGs consists of 100 DAGs uniformly sampled from the MEC of the true DAG, or the MEC itself if the MEC has size less than 100. In the latter case, the rest of the 100 DAGs are given by bootstrapping the observational data and using the techniques of Yang, Katcoff, and Uhler [40] to infer DAGs. In evaluating the finite-sample objective, there is some variance since we observe samples with noise. For evaluating methods on Objective 1 as in Figure 2(d), we average over 10 repeats. For SSG-B, which makes use of the objective for greedily selecting interventions, we approximate the objective by 10 repeats also. In Figure 2(e) we plot F1 scores for predicting the presence of edges in the true DAG. We compute the probability of each directed edge being present by

$$\mathbb{P}(u \to v) = \sum_{G \in \tilde{\mathcal{G}}} \hat{\mathbb{P}}(G)\mathbb{1}((u \to v) \in G)$$

where $\tilde{\mathcal{G}}$ is the set of bootstrapped DAGs and $\hat{\mathbb{P}}$ is the posterior after collecting samples from interventions. A predicted graph is then estimated by thresholding the edge probabilities. We select the threshold that gives the graph with maximum F1 score, and then plot the F1 score.

Similarly, we compute the weighted average (across the posterior) of structural hamming distances (SHDs) between graphs in the set of bootstrapped DAGs and the true DAG. This gives results similar to those of when we plot F1 score, as shown in Figure 9. Methods selecting the most informative

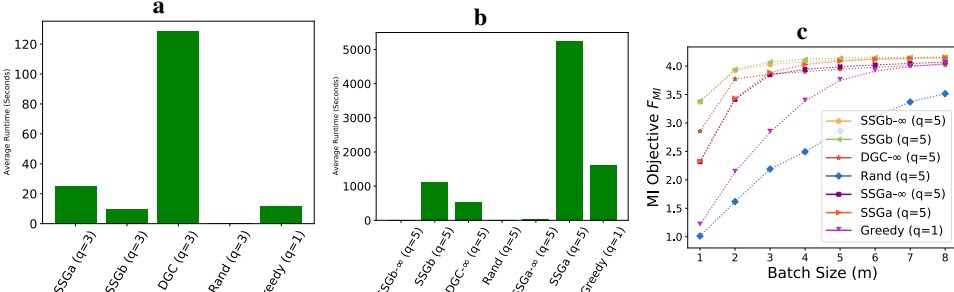

Figure 10: **a)** We demonstrate the runtime for selecting a batch of 5 interventions of size $q = 3$ with each algorithm in the infinite sample setting. This is on ER $0.1$ graphs with $p = 40$ nodes. We see that DGC is slower than other approaches but not impractical, whilst out of the SSG methods, the graph-sensitive separating system construction results in faster runtimes. **b)** We show that for selecting $8$ interventions of size $q = 5$, the infinite sample approximation methods we give result in substantially improved runtimes compared to methods that use the finite sample objective. This is again on the ER $0.1$ graphs but in the finite sample setting. **c)** We replicate Figure 2(d) but include the SSG algorithms based on the graph agnostic separating system construction. This separating system construction does not perform as well as the other algorithms we propose.

interventions will have lower mean SHD because they will decrease the posterior probability of graphs distant from the true DAG.

In Figure 10(a-b) we compare the average runtimes of our algorithms running on the same hardware in a computing cluster. We do this for both the infinite and finite sample case. We see that although the DGC algorithm is slower than alternative approach on infinite samples, in finite samples it has a faster runtime than approaches that use the true finite sample objective. We also note that SSG-B has faster runtimes than SSG-A, likely because the graph agnostic separating system construction returns larger separating systems. Whilst we aimed to maximize performance in terms of selecting the most informative experiments, these methods can be made faster at the expense of achieving lower objective values. For SSG we could take fewer gradient steps, and for DGC we could use a smaller set of sampled DAGs to approximate the objective.

In Figure 10(c) we include plot performance in finite samples of the graph agnostic separating system for SSG and see that its performance is lower than our alternative approaches.

### A.14 Dream 3 Network Experiments

In Figure 11(a–d) we give the results of DREAM3 networks not included in the main paper. For each, we record the number of edges oriented in the true DAG by each method, averaging over 5 repeats. Again, we work in the infinite samples per intervention setting. In Figure 11(e–f) we give illustrations of the ground truth DAG associated with two of the DREAM3 networks (images generated using software from Marbach et al. [24]). We note that these networks differ from typical Erdös-Renyí random graphs in that they have a few nodes with many connections, and many nodes with sparser connections.

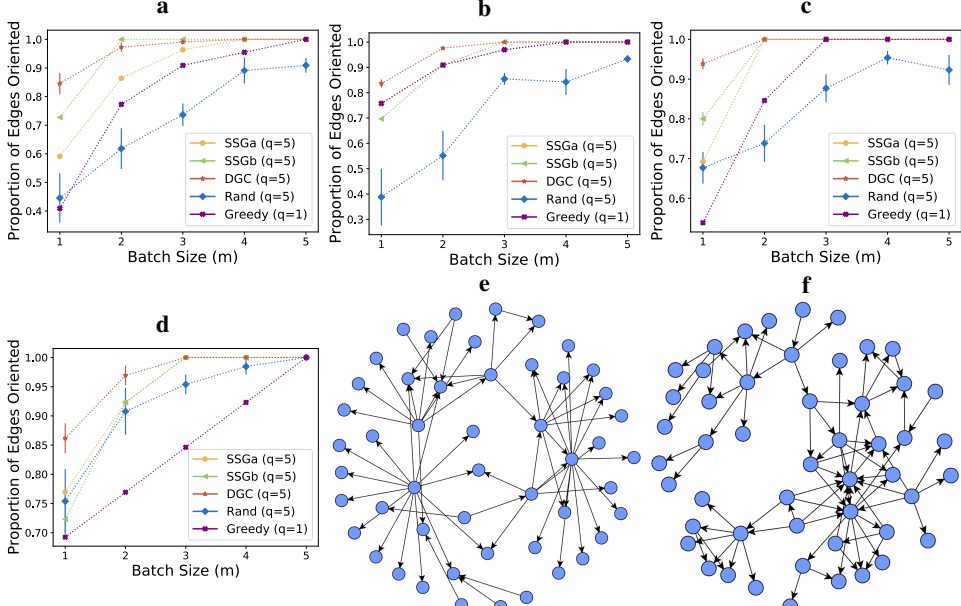

Figure 11: **a–d)** The results for experiments identical to those in Figure 2 for the $p = 50$ networks "Ecoli1", "Ecoli2", "Yeast2" and "Yeast3" from Marbach et al. [24] respectively. For an equal number of interventions and $q = 5$, our methods in general orient more edges than both random and single-perturbation greedy interventions. **e, f)** The ground truth network for "Ecoli1" and "Yeast1" respectively.