# OpenReview forum: "Near-Optimal Multi-Perturbation Experimental Design for Causal Structure Learning"
_NeurIPS.cc/2021/Conference — NeurIPS 2021 Poster_

### Official Review · Reviewer_1uN3 · 2021-06-29

**Rating:** 6
**Confidence:** 3

**Summary:**

The paper considers a challenging problem of designing multi-perturbation experiments for causal discovery. They focus on two objective functions and develop one algorithm for each objective. The approximation accuracy is studied by showing submodularity of the proposed algorithms. Empirical studies are used to demonstrate the superiority of the proposed algorithms over random design and single-perturbation design.

**Limitations And Societal Impact:**

Yes, they are mentioned.

**Main Review:**

 The major strengths of this paper are:
1.	Multiple rather than individual interventions are considered.
2.	The theoretical properties of the proposed approximated algorithms are studied through set function submodularity.

The major weaknesses are:
1.	The theoretical property is only asymptotically correct with infinite sample size. Finite sample property is not clear.
2.	Assumption on causal sufficiency is strong. One major advantage of using experimental data for causal discovery is weaker causal assumption. One critical assumption that people often make for observational data is causal sufficiency. However, in this paper, despite experimental data are generated, causal sufficiency is still required. Since it is very difficult to check its validity, it may limit the practical utility of the proposed method.

======update=====

I keep my original score.

**Time Spent Reviewing:**

5

---

> ### Author Response · Authors · 2021-08-09
> **Response to Reviewer 1uN3**
>
> Thank you very much for your comments.
>
> We agree with the limitations presented and believe these are also conveyed clearly in the paper. In terms of practicality, our assumptions are likely reasonable for applications in single-cell biology. Here we can obtain large sample sizes per intervention and measure an expression level for every gene, making the causal sufficiency assumption plausible. A goal for future work would be to relax these assumptions.

---

### Official Review · Reviewer_cWNt · 2021-07-16

**Rating:** 7
**Confidence:** 3

**Summary:**

The paper studies experimental design for causal structure learning.
Unlike existing works that focus on experiments with single
interventions, it focuses on the experiments that can possibly
simulnatenously intervene on multiple variables. The search space
becomes much larger because the power set of all variables has
exponential cardinality. The paper establish submodularity properties
for a two optimization objectives, which allows them to employ/adapt
existing variants of greedy algorithm to perform active causal
structure learning. The paper also establish theoretical guarantees
that shows the algorithm can achieve a constant factor of the optimal
objective value with the greedy algorithms.

**Limitations And Societal Impact:**

Yes the paper discusses the limitations in Sec 5.

**Main Review:**

The paper studies experimental design for causal structure learning.
Unlike existing works that focus on experiments with single
interventions, it focuses on the experiments that can possibly
simulnatenously intervene on multiple variables. The search space
becomes much larger because the power set of all variables has
exponential cardinality. The paper establish submodularity properties
for a two optimization objectives, which allows them to employ/adapt
existing variants of greedy algorithm to perform active causal
structure learning. The paper also establish theoretical guarantees
that shows the algorithm can achieve a constant factor of the optimal
objective value with the greedy algorithms.

The paper studies an original and very important scientific problem,
which emerges due to recent advances in gene-related studies.
Moreover, performing an efficient search over power set is a relevant
active learning question in many applications apart from causal
structure learning. It is well understood that submodularity of
objectives can facilitate this search using greedy algorithms. The key
observation is that some objectives in causal structure learning, in
particular, the mutual information and the edge-orientation objectives
indeed can exhibit such submodularity properties, which make this task
a possible task (at least theoretically).

Both the theoretical results and the empirical results in the paper
make sense. However, I have a few major questions.

One is about the choice of optimization objectives for mutual
information and edge orientations. I understand that these objectives
are chosen because of their submodularity-related properties. Are
there other reasons to focus on these objectives beyond submodularity
considerations? Suppose for a second we are only interested in single
node interventions. I wonder if they would still be the best/ideal
objectives for the task of structure learning or edge orientation.

The second question is about the actual gain of considering the
multiple-perturbation setting. While the theorems provide constant
approximation guarantees of the algorithm, I wonder if the paper could
elaborate on the related guarantee of single-node interventions. In
particular, one would wonder that, though we need fewer optimal
multiple-perturbation as in Figure 1, would it take more iterations
(applications of the Meek rules) to locate these optimal
multiple-perturbation? In other words, taking together the cost of
computing optimal multiple-perturbations and the gain of needing fewer
multiple-perturbations for learning, how much actual gain one would
obtain from considering multiple-perturbations? The paper would
benefit from such a discussion to solidify the motivation to consider
multiple-perturbations.

Finally, I am a little confused by the ``infinite sample'' nature of
the objective 2. Why would one be able to perform experiments with
infinite sample? How is the empirical studies done to achieve infinite
samples?


**Time Spent Reviewing:**

10

---

> ### Author Response · Authors · 2021-08-09
> **Response to Reviewer cWNt**
>
> Thank you very much for your comments.
>
> **Choice of objectives.** The edge-orienting objective is an intuitive objective when in the infinite sample case. Maximizing the objective completely leads to unique identification of the causal structure. Agrawal et al. [3] introduce the MI objective for causal structure learning in the finite sample setting, and show that it has a consistency property for uniquely identifying the true causal structure when optimized (Theorem 3.4 in their paper). The exact objective that makes most sense will depend on what the use of the structure estimate will be in downstream prediction or decision tasks.  In the experiments, Figure 2e shows that optimizing the MI objective corresponds well with downstream performance on the task of identifying the presence and direction of all edges in the causal structure.
>
> **Comparing the guarantee to the guarantee for single node interventions.** We will include a comparison in Section 4.1 of the camera-ready version. In particular, we will compare the guarantee in Theorem 2 with the bound in Ghassami et al. [12]. For single node interventions, the constant factor in the guarantee is more favourable. However, said guarantee is with respect to the optimal single perturbation intervention batch, whereas our bound is with respect to the best multi-perturbation intervention batch that satisfies the intervention size constraint. We discuss in the introduction, with more detail in the supplement, that the optimal multi-perturbation intervention could achieve up to a factor of $q$ times greater objective value.
>
> **Why would one be able to perform experiments with infinite samples per unique intervention?** This assumption leads to a tractable theory, but is of course not strictly true in practice. It may be a good approximation of reality in the setting where we study a system with little noise in the structural equation models. It may also be a good approximation when we are constrained to selecting few unique interventions, but can get many samples of each of these. The finite sample experiments we perform demonstrate that the theoretically grounded algorithms we develop based upon this assumption lead to practical algorithms under finite samples.
>
> **How are the empirical studies done to achieve infinite samples?** In the infinite sample setting, there are graphical criteria that can be used to determine which edges in an MEC are oriented, given the ground truth DAG and a batch of interventions [24].

---

> > ### Comment · Reviewer_cWNt · 2021-08-31
> > **Thank you for your response**
> >
> > Thank you for your response. My (positive) evaluation of the paper stays.

---

### Official Review · Reviewer_ZpMg · 2021-07-16

**Rating:** 8
**Confidence:** 4

**Summary:**

This paper tackles the problem of causal structure learning from interventional data. Specifically, they seek a series of interventional experiments which can rapidly identify the correct causal graph, or reduce some measure of uncertainty about the true causal structure. In contrast to previous work, this paper considers the more general setting in which interventions can be made on more than one variable simultaneously (within one experiment). To select an optimal sequence of subsets on which to intervene, the authors begin by discussing 3 objectives functions that could be used to select “good” sets of interventions: two based on MI, one based on orienting edges. All objectives are grounded in existing literature. The authors propose two different algorithms to actually solve the experimental design problem. First, they propose Double Greedy Continuous. As the name implies, this method is greedy both with respect to adding interventions to the experiment sequence and with respect to adding variables to intervene on in a single interventional experiment. The authors provide strong theoretical guarantees for this method with respect to their edge-orienting objective. For the MI objective, a suitable sub-modularity property does not hold for double greedy. Instead, the authors propose a new algorithm, Separating Systems Greedy. This is greedy with respect to adding interventional experiments, but to choose sets of variables to intervene on, this algorithm uses an exhaustive search over a separating system. The authors prove guarantees relating this algorithm to their infinite data MI objective.

**Limitations And Societal Impact:**

I believe the authors have dealt sufficiently with societal impact. For discussion on limitations, see the main review.

**Main Review:**

Originality: This paper tackles the problem of finding sequences of multi-variable interventional experiments to do causal structure learning. Few, if any, previous works have studied this exact form of the causal structure learning. The paper is strictly more general than the previous work of Agrawal et al., 2019 and Ghassami et al., 2018 in terms of the problem addressed.

The work proposes two novel algorithms, along with a number of new theoretical insights relevant to this problem and their proposed methods. The only limitation here is that the actual algorithms suggested (particularly double greedy) are natural extensions of existing work. However, this should not overly detract from the value of this paper.

When discussing MI as an objective for experimental design, it could be nice to cite e.g. Lindley, 1956 ‘On a measure of the information provided by an experiment’. Also, you define $\tilde{U}_{M.I.}(y, \xi; D)$ as the mutual information between y and the posterior over G. Typically the mutual information would imply an expectation over y. Perhaps it is the information gain/KL divergence from prior to posterior on G.

Quality: The submission is of high quality. The approach taken is sound, logical and well-motivated. The paper flows logically with the authors defining and breaking down the problem, before proposing solution and then analysing them theoretically. The suggested methods are well supported by theory. I haven’t checked all proofs in the appendix, but the parts I checked were good. Experimentally, the authors demonstrate performance gains over the key baseline of single-variable intervention experiments, as well as comparing their different methods. A few things that could be improved
Further discussions of limitations and trade-offs. For example, are there cases when single-perturbation experiments might be expected to suffice? What considerations should a practitioner use to decide between DSC and SSG? For example, in 264-265, it is explained that under certain conditions SSG-B may not improve with q. This idea could be bumped up to a more general discussion before/after the experiments. Similarly, it could help to simply restate the bounds on the separating system sizes from Shanmugam et al. and Lindgren et al.
Can we say anything theoretically about Objective 1 / the finite sample case?

Clarity: the paper is well written with a number of helpful illustrative diagrams (very nice for any DAG paper). The text on the images of Fig 2 are too small.

Significance: This paper tackles a significant problem, making major strides in what is achievable or considered in the literature. There are many applications for causal structure learning, gene regulatory networks, but also many others.

**Time Spent Reviewing:**

6

---

> ### Author Response · Authors · 2021-08-09
> **Response to Reviewer ZpMg**
>
> Thank you very much for your comments.
>
> **When might single variable perturbations suffice?**
> We did not find general graph structures during the experiments where single perturbation experiments performed very similarly to multi-perturbation interventions. The only exception is the trivial example of structures where the MEC is very close to a star, where one single perturbation intervention can orient most or all edges.
>
> **"Can we say anything theoretically about Objective 1 / the finite sample case?"** Currently there is not an accompanying finite sample guarantee for Objective 1. This is a direction for future work.
>
> **What considerations should a practitioner use to decide between DSC and SSG?**
> The experiments on adversarial examples and the constant factor guarantee for DSC on Objective 3 suggest that DSC is likely the best choice of algorithm when we want to ensure robustness against unfavourable MEC structures.

---

### Official Review · Reviewer_ftg7 · 2021-07-16

**Rating:** 6
**Confidence:** 4

**Summary:**

The authors propose a greedy intervention design algorithm to optimize a collection of objectives where they show and leverage submodularity properties of these objectives, making use of several existing works in the causal discovery literature.


**Limitations And Societal Impact:**

yes.

**Main Review:**

The paper seems to make interesting connections to continuous optimization through submodularity. The main issue I currently have is that it is not clear to me how the number of Meek rule applications appears in this continuous optimization framework. Please comment on this. I will be happy to update my score based on this and other major comments below (positioning of the paper and additional experiments)

On positioning and presentation:
The way the paper positions itself in terms of novelty is a bit misleading especially the phrase "largely unexplored" in the abstract. There are indeed several papers on experimental design including multi-perturbation. I believe the authors should be on point and present their result as a follow-up of the style of [3,12] rather than claiming there isn't much work on multi-perturbation interventional design.

Similarly the claim "algorithmically challenging because it leads to an exponentially large search space" is also specific to the line of work that this paper follows upon and should not be presented as a general challenge. This overall narrative is a bit misleading and makes it hard to understand and appreciate the paper's contributions in relation to the related work.

Please clarify in the introduction that you assume no latent variables.

The faithfulness assumed should be interventional faithfulness of Hauser and Buhlmann'12 and not only faithfulness unless I am missing something, please clarify.

Details:
Number of unoriented edges are directly related to the size of equivalence class through the essential graph. This relation should be explained in more detail to flesh out the distinction between Obj2 and Obj3.

Notation of Obj3 is a bit confusing since there is no mention of \xi prime - already conducted experiments. Proof of lemma 1 mentions this is implicit in R but please add it as subscript to make it clear.

Proposition 3 is missing the condition that \xi1 \subseteq \xi2

Proposition 4. I believe the authors should check the details of Hauser and Buhlmann to see if there are results they can pull in. I believe Proposition 4 or a version of it must appear there in the equivalence class characterization.

Lemma 2 is surprising and seems important enough to be called a theorem.

It is not clear how the runtime of [25] is converted to Meek rules. This point should be clarified by the authors in the rebuttal and in the paper. Meek rules are just a subroutine to orient edges of an essential graph and why it appears as a parameter in a continuous optimization framework needs to be explained in much greater detail. (used crucially in Theorem 1, Theorem 2 and Theorem 3)

Experiments:
A directly related method in the infinite-sample regime is the adaptive algorithm of Shanmugam et al. [30]. Please add comparisons with this algorithm. ​This is a method outside the line of work this paper is following up on and it would be good to have a comparison with a method that is not directly related.

Post-Rebuttal:
The authors clarified my concerns about how Meek rules appear in the theorem statements. They have also explained the difference between the batch setting they are interested in and the adaptive setting of [30]. Based on our exchange, I am increasing my score.

**Time Spent Reviewing:**

4

---

> ### Author Response · Authors · 2021-08-09
> **Response to Reviewer ftg7**
>
> Thank you very much for your comments.
>
> **The relationship between Meek rule applications and the continuous optimization framework**
>
> For the purpose of explanation we will ignore the slightly more general verson of Objective 3 stated between lines 178-179 and instead just work with the one listed in Equation 3. $F_{EO}(\xi) = \frac{1}{\mathcal{G}} \sum_{G \in \mathcal{G}}|R(\xi, G)|$ can be written as a sum of function calls that each depend on just a single member of the equivalence class. By number of applications of the Meek rules, we mean the number of times function $R$ must be evaluated for a $(\xi, G)$ pair.
> The dependence upon time to evaluate $R$ comes about as follows. In Algorithm 1, we call the subroutine NMSCG. NMSCG estimates the gradient of the multilinear extension of $F^{\xi}_{EO}$ multiple times during each call to the subroutine. This gradient is estimated by evaluating $R$ at two different intervention values for a randomly drawn graph from the MEC (the equation right before line 192).
>
> In the camera-ready version, we will reword what is meant by an application of the Meek rules in this context, and refer to it as number of evaluations of function $R$. We will add a sentence to explain that the runtime of a single evaluation of $R$ is polynomial in the number of nodes, since it involves computing the Meek rules for a specific graph and intervention until no more edges can be oriented (described on line 110).
>
> **Comparing empirically to an adaptive algorithm such as Shanmugam et al. [30].** To our understanding, Algorithm 1 in the referenced paper is not designed for the batched setting. It allows for the selection of sequential experiments when the batch size is 1. It is not clear how to directly compare this algorithm with ours, since our target domain is batched experiments and our evaluation is in the batched setting. We would appreciate it if the reviewer could clarify what they mean by such a comparison so that we could comment further.
>
> **Contribution.** We will revise the wording of the abstract to make it clear that this is the first work to consider multi-perturbation interventions, and the associated combinatorial challenges, specifically **in the batched causal discovery setting**.

---

> > ### Comment · Reviewer_ftg7 · 2021-08-23
> > **rebuttal**
> >
> > Hi! Thank you for your responses.
> >
> > I am still quite confused about the Meek rule statements and would greatly appreciate it if the authors could clarify further. More specifically, I think there needs to be a Proof of Theorem 1 in the appendix of the paper. Currently, authors only cite [25] however this doesn't help the reader understand the role of Meek rules in this continuous optimization framework. I checked [25] but was unable to make this connection myself. Therefore, I would really appreciate any further input on this.
> >
> > I would also appreciate it if the authors could respond to some of the other comments in my initial review, such as requiring interventional faithfulness, clarifying that there assumed to be no latents, and the relation of Lemma 4 to Hauser and Buhlmann's characterization.
> >
> > On experiments and comparison w/ [30]:
> > "It allows for the selection of sequential experiments when the batch size is 1"
> > Based on my understanding, their algorithm identifies an experiment of size-k for the given k at each step, i.e., allows batch size to be k>1 (see lines 13 and 15 of Algorithm 1 in [30]). The intervention is then performed concurrently on all k nodes in a single experiment, similar to the setting in this paper. However, the current paper focuses on the non-adaptive setting, where the experiments are designed at the beginning and are conducted in parallel, whereas the aforementioned algorithm is adaptive. It is, therefore, not essential to have such a comparison. It would still be interesting to see this experiment if the authors choose to do so in the camera-ready, which would really demonstrate whether adaptivity helps in practice and if so by how much.

---

> > > ### Author Response · Authors · 2021-08-25
> > > **response @ftg7**
> > >
> > > **Theorem 1 and Meek rules** : Theorem 1 corresponds to corollary 14 in [25]. We will again work with Objective 3 in the form that it is presented in line 109 of our paper. Recall that the purpose of this theorem is to have an optimality guarantee on Objective 3 for selecting a single multi-perturbation intervention $I$ and adding it to our existing batch $\xi$.
> > >
> > > Corollary 14 in [25] states the following. Let $f(S) = \\mathbb{E} [\\tilde{f}(S, z)]$ (expectation taken over $z \\sim P$) be a submodular set function (using notation from line 33 of [25]). For us $z$ is a graph and $P$ is the uniform distribution over the MEC. $S$ is the experiment to be added to the batch, which we call $I$.  $\\tilde{f}$ is the function $\\tilde{f}(I, G) = | R(\\xi \\cup \\{I\\}, G) |$ and $f$ is our objective function $F_{EO}^{\xi}$.  The corollary states that, using NMSCG and rounding, maximizing $f$ requires $\mathcal{O}(n^{5/3} / \epsilon^3)$ evaluations of function $\tilde{f}$ in order to reach a $(1/e) OPT - \epsilon$ solution (in expectation over randomness in the rounding procedure). Here $n$ is the dimension of the input to $f$, so in our case the number of nodes $p$ in the graph, since we can choose whether to intervene or not on each in a given intervention $I$.
> > >
> > > In our paper, by "applications of Meek rules", we mean number of evaluations of $R$. As mentioned in our previous response we will change this wording in the camera-ready version. In the updated version there will be no mention of Meek rules in the theorem statement. We had this wording before because the process of computing Meek rules, until no more edges can be oriented, is the bottleneck (in terms of time) for computing the function $R$ for a single $G, \xi$ pair. In the paragraph beginning on line 110, we describe the relationship between computing $R$ and the Meek rules.
> > >
> > > In the camera-ready version Theorem 1 will read as follows:
> > >
> > > Let $I^*$ be the maximizer of $F_{EO}^{\xi}$. NMSCG with pipage rounding, after $\mathcal{O} \left( p^{5/2}/\epsilon^3 \right)$ function evaluations of $R$, achieves a solution $I$ such that
> > > $$\\mathbb{E}\left[F_{EO}^{\xi}(I)\right] \geq \frac{1}{e} F_{EO}^{\xi}(I^*) - \epsilon.$$
> > >
> > > We will then mention elsewhere that $R$ can be evaluated in polynomial time, because the two steps described from line 110 onward can be computed in polynomial time. We want to emphasize that this is simply a rewording for the purpose of clarity and is not a change in the results of the paper.
> > >
> > > Corollary 14 in [25] (Theorem 1 for us) does have a further condition called assumption 4 in [25], which is a statement regarding the family of constraints on the input $S$ that corollary 14 holds for. The constraint in our case, for selecting one multi-perturbation intervention, is a cardinality constraint on the number of nodes selected for the intervention, and the assumption holds trivially in this case.
> > >
> > > **Interventional faithfulness**: you are correct that we in fact assume interventional faithfulness and we will clearly specify this in the camera-ready version.
> > >
> > > **No latents**: yes we assume no latent confounders. This is stated in different words on line 83.
> > >
> > > **Hauser and Buehlmann: Proposition 4** is most related to a result in the original Meek rules paper, which we invoke several times in the proof. We have carefully checked the Hauser and Buehlmann paper and there is not an equivalent result. Theorem 18 in Hauser and Buehlmann could likely also be used to prove our result, but it is not straightforward, and the approach we took allowed us to make substantial use of the similar result in Meek's paper.
> > >
> > > **Experiments**: the confusion here is likely that by "batch size", we mean the number of experiments performed simultaneously. You are referring to batch size as the maximum number of nodes selected in each experiment. In our terminology the adaptive algorithm of [30] is an algorithm with batch size 1. We agree with your description of the algorithm in [30]. That is, the difference in the settings between our algorithms is that we perform experiments in parallel whilst they perform them sequentially. We agree: whilst the proposed experiment does not directly address the main focus of our paper, it would provide information on the value of being able to perform experiments adaptively. In the particular application most relevant to our paper, cell biology experiments, there is a high cost to starting a new batch of experiments but a relatively lower cost for adding new experiments to an existing batch. Reasons for this include experiment run-time, human labour cost to begin the setup of each batch and the use of time on equipment that is designed for performing experiments in parallel. In these cases the performance of algorithms with batch size of 1 is not of interest because such experiments would not be practical. Finally, note that our algorithm could be applied in an adaptive, batched setting. That is, to select several batches of experiments sequentially.
> > >
> > > Please let us know if you have any follow-up questions. In particular, if the relationship between the Meek rules and computation of $R$ or the results of Theorems 1 and 2 are not clear yet, we would be very happy to elaborate on this further. Thank you for your follow-up and helping us improve the paper, especially on this point.

---

> > > > ### Comment · Reviewer_ftg7 · 2021-09-01
> > > > **response**
> > > >
> > > > Hi, thank you for your response.
> > > >
> > > > I believe I understand the source of Meek rule statements now. Yes, the current wording of the theorem can even be interpreted as "a certain number of Meek rule orientations is necessary" so it would be great to remove Meek rules from the statement as suggested by the authors, but also explain how they typically create the bottleneck in terms of the number of function evaluations.
> > > >
> > > > Batch size: I understand the exact setting, I believe. Batch size of 1 corresponds to an adaptive experiment setup and a batch size of infinity would correspond to a non-adaptive or offline experimental design. Any k in-between corresponds to being able to perform k interventions in parallel, but then one needs to design the next batch of k experiments to be performed simultaneously in an adaptive manner - i.e., using the output of the previous experiment. Is my understanding correct?
> > > >
> > > > I will increase my score to reflect the clarifications. Thank you again for engaging in the discussion!

---

> > > > > ### Author Response · Authors · 2021-09-01
> > > > > **response @ftg7**
> > > > >
> > > > > Thank you very much for taking the time to discuss these details with us.
> > > > >
> > > > > When we can do $N$ total interventions, a batch size of $k=1$ is an adaptive experiment setup and a batch size of $k=N$ (rather than infinity) is a totally offline setup. Any batch size $k \in (1, N)$, exactly as you said, leads to designing a set of $k$ experiments all performed simultaneously, and the design can be based upon feedback from the previous batches.

---

### Decision · Program_Chairs · 2021-09-27

**Decision:**

Accept (Poster)

**Comment:**

This paper describes a methodology for causal structure learning while varying multiple variables simultaneously. Individually, the fields of structure learning, causal inference, and experiment design are well-studies areas of statistics and machine learning. The novelty of this work is bringing such ideas together to address causal structure learning. The paper makes interesting connections to submodularity to solve the problem.

The reviewers were supportive of acceptance of the paper and found numerous merits. First, the proposed approximation algorithm is analyzed theoretically using set function submodularity. This lays the foundation for further analysis and perhaps improvements. Second, the empirical results show that there are significant improvements to be found by sequential multi-perturbations in causal structure learning. There were some clarifying comments around so-called Meek rules. The authors provided clarification in their response and the associated adjustments to the paper seem straightforward. Given that the paper was found to be technically sound and potentially impactful for not only gene regulatory networks, but also for other problem domains, I agree with the reviewers and favor acceptance.